biomathematics/systems biology

theoretical biology, Bayesian parameter estimation, cocoa bean fermentation, kinetic modelling

**Author for correspondence:**
Marc-Thorsten Hütt
e-mail: m.huett@jacobs-university.de

# Exploring cocoa bean fermentation mechanisms by kinetic modelling

Mauricio Moreno-Zambrano, Matthias S. Ullrich and Marc-Thorsten Hütt

Department of Life Sciences and Chemistry, Jacobs University Bremen, Campus Ring 1, 28759 Bremen, Germany

MM-Z, 0000-0003-3446-8720; M-TH, 0000-0003-2221-423X

Compared with other fermentation processes in food industry, cocoa bean fermentation is uncontrolled and not standardized. A detailed mechanistic understanding can therefore be relevant for cocoa bean quality control. Starting from an existing mathematical model of cocoa bean fermentation we analyse five additional biochemical mechanisms derived from the literature. These mechanisms, when added to the baseline model either in isolation or in combination, were evaluated in terms of their capacity to describe experimental data. In total, we evaluated 32 model variants on 23 fermentation datasets. We interpret the results from two perspectives: (1) success of the potential mechanism, (2) discrimination of fermentation protocols based on estimated parameters. The former provides insight in the fermentation process itself. The latter opens an avenue towards reverse-engineering empirical conditions from model parameters. We find support for two mechanisms debated in the literature: consumption of fructose by lactic acid bacteria and production of acetic acid by yeast. Furthermore, we provide evidence that model parameters are sensitive to differences in the cultivar, temperature control and usage of steel tanks compared with wooden boxes. Our results show that mathematical modelling can provide an alternative to standard chemical fingerprinting in the interpretation of fermentation data.

## 1. Introduction

Cocoa beans from *Theobroma cacao* L. are the raw material of chocolate. Their fermentation plays a fundamental role as being responsible for eliminating undesired properties from freshly harvested beans, e.g. astringency and bitterness, besides yielding chocolate-related flavour and aroma precursor compounds [1,2]. In contrast to the highly controlled fermentation processes known from other food products, this process is conducted *in situ* at each of the producing farms in a

spontaneous form varying in both, methodology, e.g. wooden boxes, heaps and platforms [2–4], and observed microbial diversity [5].

This heterogeneity due to different fermentation methods and indigenous microbiota, leads to a plethora of studies that have qualitatively described the process, e.g. [4,5]. Among all these, sequentiality of microbial populations thriving on the beans' enclosing pulp constitutes the process dynamics with greatest acceptance [1,2,4].

In further detail, regardless of the wide range of factors that could differentiate fermentation trials, sequential succession of microbial groups during their execution can be understood as a three-phased process, where a microbial group dominates each phase during a distinct time period. In a first stage, anaerobic conditions due to the packed nature of the pulp favour the growth of yeasts that bloom as a consequence of a carbohydrate-rich environment producing mainly ethanol. Through their pectinolytic action yeasts drive to a liquefaction of the pulp. As a consequence, a drainage of pulp permits air to enter into the fermenting mass, contributing to the decline of yeast population. Under these conditions, a second stage is dominated by the growth of microaerophilic lactic acid bacteria (LAB) that at the onset of the process were reproducing at a lower rate than yeasts. Therefore, by depletion of remaining sugars from the first stage, LAB yield mainly lactic and acetic acids. At this point, after considerable drainage of pulp, a fully aerobic phase is reached. This third and final stage is characterized by an almost complete dominance of aerophilic acetic acid bacteria (AAB) that oxidize lactic acid into acetoin, and ethanol into acetic acid [1,3,4,6].

As a consequence, microbial sequentiality during the fermentation has served as the basis in formulating a few mathematical approaches for its quantitative description [7–10]. Among these, we previously proposed a mathematical model of ordinary differential equations (ODEs), which served us as baseline here [9]. The main sub-processes implemented in this study focus on the activity of major microbial groups, namely yeasts (Y), LAB and AAB. As a result, we developed a successful model based on well-known regulatory assumptions: in a first instance, Y come into play by converting glucose (Glc) and fructose (Fru) into ethanol (EtOH). Concomitantly, LAB consumes Glc leaving as products lactic acid (LA) and acetic acid (Ac). Finally, AAB takes over the last phase of fermentation by oxidizing EtOH and LA into Ac [9].

Beyond these main components, more regulatory mechanisms have been mentioned across experimental studies that could bring more insight into the dynamics of cocoa bean fermentation. Among these, we here put special emphasis in five phenomena (see detailed references in Materials and methods, below): (i) decrease of product metabolites by physical causes, (ii) consumption of Fru by LAB, (iii) production of Ac by Y, (iv) consumption of LA by Y, and (v) over-oxidation of Ac by AAB.

Along these lines, we were able to assess the plausibility of stand-alone and simultaneous occurrence of these mechanisms when added to our baseline model and to identify systematic differences of fermentation features by applying classification methods over their resulting vectors of parameter estimates. Our key questions are: (i) Which model variants describe the experimental data better than the baseline model? (ii) For which model can parameter differences be related to differences in the fermentation process?

# 2. Material and methods

## 2.1. Identification and processing of experimental data

A literature survey concerning cocoa bean fermentation trials was performed with the purpose of gathering experimental data. Reported trials considered in this study were papers published between 2000 and 2019. As inclusion criteria, only English-written works with time series of minimum five observations for metabolites Glc, Fru, EtOH, LA and Ac, besides total population counts of Y, LAB and AAB were included.

In all cases, population growth of Y, LAB and AAB were transformed from log base 10 of colony forming units ($\log_{10}$(CFU)) to milligrams of microbial group (MG) per gram of pulp (mg(MG)g(pulp)$^{-1}$) as these are the units in which most kinetic single-strained microbial growth studies report their dynamics as well as their dependent constants, i.e. specific maximum growth and mortality rates, and yield coefficients. Moreover, with the purpose of facilitating the estimation of models' parameters by avoiding numerical issues during their calibration, all time series were scaled by dividing each observation by its own maximum value. Hence, obtained parameter estimates were rescaled to their original units by using simple transformations for their further comparison with previously reported values for single-strained microbial studies (see electronic supplementary material, S1) [9].

For this current research, distinct trials were given a code name based on country of origin and fermentation method. A complete detail of data included in this research where at least one model variation successfully fit it (see following sections for their explanation) is shown in table 1. For a comprehensive list of all data initially considered see electronic supplementary material, S2.

## 2.2. Formulation of candidate models

Starting from the baseline model from [9], we implemented five regulatory mechanisms that have been reported or hypothesized in multiple studies. In the following paragraphs, the baseline model will be described and proposed mechanisms reasoning will be presented conforming what we considered their likeliness of occurrence as (i) decay of fermentation's products, (ii) consumption of Fru by LAB, (iii) production of Ac by Y, (iv) consumption of LA by Y, and (v) over-oxidation of Ac by AAB.

### 2.2.1. Baseline model

As baseline, we used our previously developed model [9] that consist of eight ODEs describing the dynamics of metabolites: Glc, Fru, EtOH, LA and Ac, besides microbial groups: Y, LAB and AAB. Both metabolites and microbial groups are interdependent in the dynamic process by means of growth and mortality rates of the latter (figure 1a). Monod [23] and Contois [24] type equations were employed to describe the growth rates of microbial groups. Growth rates $v_1$ and $v_2$ of Y on Glc and Fru, respectively, as well as growth rates $v_3$ of LAB on Glc and $v_4$ of AAB on EtOH, correspond to Monod equations, while the growth of AAB on LA, $v_5$, corresponds to a Contois term. Mortality rates of Y, LAB and AAB were modelled as Chick–Watson equations [25] by considering second- and third-order death kinetics, as shown in table 2.

The model contains 24 parameters: five maximum specific growth rates, five substrate saturation constants, three mortality rate constants and 11 yield coefficients as depicted in table 2 and the following equations:

$$\frac{d[\text{Glc}]}{dt} = -Y_{\text{Glc}|\text{Y}} v_1 - Y_{\text{Glc}|\text{LAB}} v_3 \tag{2.1}$$

$$\frac{d[\text{Fru}]}{dt} = -Y_{\text{Fru}|\text{Y}} v_2 \tag{2.2}$$

$$\frac{d[\text{EtOH}]}{dt} = Y_{\text{EtOH}|\text{Y}}^{\text{Glc}} v_1 + Y_{\text{EtOH}|\text{Y}}^{\text{Fru}} v_2 - Y_{\text{EtOH}|\text{AAB}} v_4 \tag{2.3}$$

$$\frac{d[\text{LA}]}{dt} = Y_{\text{LA}|\text{LAB}}^{\text{Glc}} v_3 - Y_{\text{LA}|\text{AAB}} v_5 \tag{2.4}$$

$$\frac{d[\text{Ac}]}{dt} = Y_{\text{Ac}|\text{LAB}}^{\text{Glc}} v_3 + Y_{\text{Ac}|\text{AAB}}^{\text{EtOH}} v_4 + Y_{\text{Ac}|\text{AAB}}^{\text{LA}} v_5 \tag{2.5}$$

$$\frac{d[\text{Y}]}{dt} = v_1 + v_2 - v_6 \tag{2.6}$$

$$\frac{d[\text{LAB}]}{dt} = v_3 - v_7 \tag{2.7}$$

and

$$\frac{d[\text{AAB}]}{dt} = v_4 + v_5 - v_8. \tag{2.8}$$

In regard to our proposed mechanisms, their inclusion into the baseline model is conducted by adding extra growth and mortality rates, as well as linear terms when needed (table 2). For a deeper look into their mathematical formulation, see electronic supplementary material, S3.

### 2.2.2. Mechanism 1: decay of fermentation products

Mechanism 1 (M1) is based on concentration decline of product metabolites at later stages of fermentation that has been hypothesized as a consequence of both physical and biological constraints. Here, we will take into account the first group only. Among these, volatile compounds (e.g. EtOH and Ac) might decrease as a result of evaporation and leakage of fermentation sweating [3,6,11,13]. Regarding non-volatile compounds (e.g. LA), the widely described diffusion process of metabolites from the pulp into the cocoa bean, might also play an important role in their reduction [4,6,26].

**Table 1.** Considered data sources.

| reference | year | country | cultivar | method | trial | code | turning | Ctrl. Temp. |
|---|---|---|---|---|---|---|---|---|
| Camu et al. [11] | 2007 | Ghana | Criollo/Forastero | heap | heap 5 | ghhp1 | ✗ | ✗ |
| Lagunes Gálvez et al. [12] | 2007 | Dominican Republic | Trinitario | wooden box | NA | dowb1 | ✓ | ✗ |
| Camu et al. [13] | 2008 | Ghana | NA* | heap | heap 10 | ghhp2 | ✓ | ✗ |
| | | | | | heap 11 | ghhp3 | ✗ | ✗ |
| | | | | | heap 12 | ghhp4 | ✓ | ✗ |
| | | | | | heap 13 | ghhp5 | ✗ | ✗ |
| Papalexandratou et al. [14] | 2011 | Brazil | Criollo/Forastero | wooden box | box 1 | brwb1 | ✓ | ✗ |
| | | | | | box 2 | brwb2 | ✓ | ✗ |
| Papalexandratou et al. [15] | 2011 | Ecuador | Nacional/Trinitario | platform | P1 | ecpt1 | ✗ | ✗ |
| | | | | | P2 | ecpt2 | ✗ | ✗ |
| | | | | wooden box | B1 | ecwb1 | ✓ | ✗ |
| | | | | | B2 | ecwb2 | ✓ | ✗ |
| Pereira et al. [16] | 2012 | Brazil | NA* | plastic box | PC | brpb1 | ✓ | ✓ |
| | | | | stainless tank | ST | brst1 | ✓ | ✓ |
| Pereira et al. [17] | 2013 | Brazil | Mixed hybrids* | wooden box | WB1 | brwb3 | ✓ | ✗ |
| | | | | | WB2 | brwb4 | ✓ | ✗ |
| | | | | stainless tank | SST | brst2 | ✓ | ✗ |
| Moreira et al. [18] | 2013 | Brazil | PH16 | wooden box | PH16 | brwb7 | NA | ✗ |
| Papalexandratou et al. [19] | 2013 | Malaysia | Mixed hybrids | wooden box | box 2 | mywb3 | ✓ | ✗ |
| Romanens et al. [20] | 2018 | Honduras | IMC-67, UF-29, UF-668 | wooden box | OF-F | hnwb1 | ✓ | ✗ |
| Lee et al. [21]† | 2019 | Ecuador | Criollo | plastic box | NA | ecpb1 | NA | ✓ |
| Papalexandratou et al. [22] | 2019 | Nicaragua | Nugu/O'payo | wooden box | NUGU | niwb1 | ✓ | ✗ |
| | | | | | O'PAYO | niwb2 | ✓ | ✗ |

Only fermentation trials that were successfully described by at least one model iteration (MI) are listed. Author, year of publication, cocoa country of origin, cocoa cultivar, used methodology, code name given in the original trial, recoded given name in this research, turning of the fermenting mass and controlled temperature are shown.

*Unidentified cultivars used by Camu et al. [13], Pereira et al. [16] and Pereira et al. [17] were coded as un1, un2 and un3, respectively, for further PCA.

†Simulated fermentation.

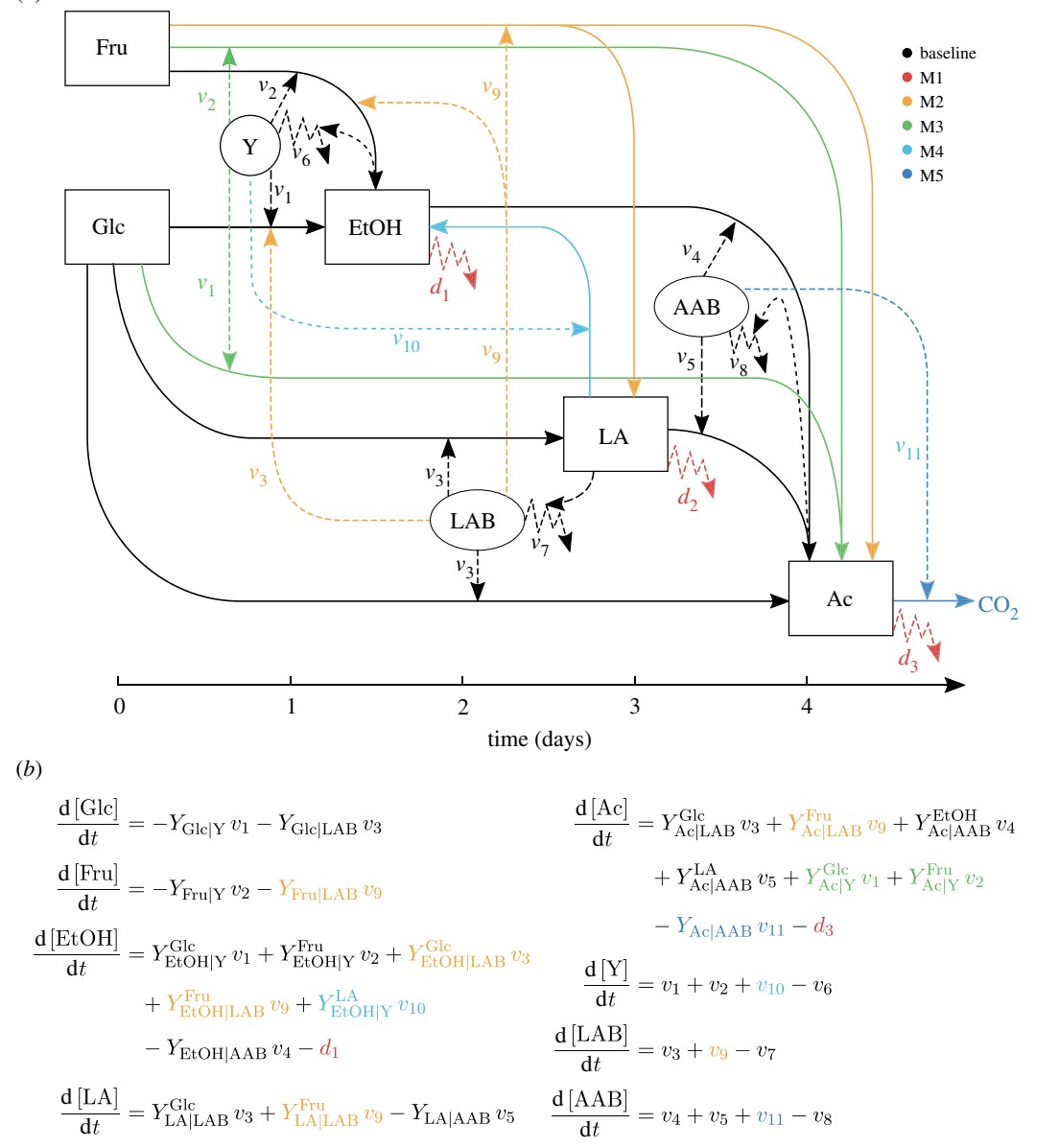

**Figure 1.** Summary of models iterations. (*a*) Network diagram of mechanisms over baseline model. Microbial groups: yeast (Y), lactic acid bacteria (LAB) and acetic acid bacteria (AAB) are represented as circles. Metabolites: glucose (Glc), fructose (Fru), ethanol (EtOH), lactic acid (LA) and acetic acid (Ac) are represented as squares. The growth rates of Y on Glc ($v_1$), Fru ($v_2$) and LA ($v_{10}$), of LAB on Glc ($v_3$) and Fru ($v_9$), and of AAB on EtOH ($v_4$), LA ($v_5$) and Ac ($v_{11}$) are represented as straight dashed arrows. The mortality rates of Y ($v_6$), LAB ($v_7$) and AAB ($v_8$) are represented as zigzag dashed arrows as the decay rates of EtOH ($d_1$), LA ($d_2$) and Ac ($d_3$). Straight dashed arrows pointing from products to mortality rates represent product influence on mortality rates. Solid straight arrows show the direction in which the conversion of metabolites occur. Baseline model by Moreno-Zambrano *et al.* [9] comprehends mechanisms depicted in black (circle). (*b*) Representation of full model with mechanisms M1, M2, M3, M4 and M5 together. M1 (red circle), encompasses losses of EtOH, LA and Ac. M2 (orange circle), involves conversion of Glc into EtOH, and Fru into EtOH, LA and Ac by LAB. M3 (green circle), comprises conversion of Glc and Fru into Ac by Y. M4 (light blue circle), refers to conversion of LA into EtOH by Y. M5 (dark blue circle), represents over-oxidation of Ac by AAB.

## 2.2.3. Mechanism 2: consumption of Fru by LAB

Opposed to our original approach of modelling LAB growth exclusively based on Glc uptake [9], mechanism 2 (M2) takes into account obligatory and facultatively heterofermentative species which are capable of using Glc and Fru as carbon sources (e.g. *Limosilactobacillus fermentum* and *Lactiplantibacillus plantarum*, respectively) with an accompanying production of EtOH besides LA and Ac [3,4,12,22,27,28].

**Table 2.** Growth, mortality and decay rates for cocoa bean fermentation models.

| growth rate equation | mortality rate equation | decay rate equation |
|---|---|---|
| $v_1 = \frac{\mu_{max}^{Y_{Glc}} [Glc]}{[Glc]+K_{Glc}^{Y}} [Y]$ | | |
| $v_2 = \frac{\mu_{max}^{Y_{Fru}} [Fru]}{[Fru]+K_{Fru}^{Y}} [Y]$ | $v_6 = k_Y [Y] [EtOH]$ | $d_1 = b_{EtOH} [EtOH]$ |
| $v_{10} = \frac{\mu_{max}^{Y_{LA}} [LA]}{[LA]+K_{LA}^{Y}} [Y]$ | | |
| $v_3 = \frac{\mu_{max}^{LAB_{Glc}} [Glc]}{[Glc]+K_{Glc}^{LAB}} [LAB]$ | $v_7 = k_{LAB} [LAB] [LA]$ | $d_2 = b_{LA} [LA]$ |
| $v_9 = \frac{\mu_{max}^{LAB_{Fru}} [Fru]}{[Fru]+K_{Fru}^{LAB}} [LAB]$ | | |
| $v_4 = \frac{\mu_{max}^{AAB_{EtOH}} [EtOH]}{[EtOH]+K_{EtOH}^{AAB}} [AAB]$ | | |
| $v_5 = \frac{\mu_{max}^{AAB_{LA}} [LA]}{[LA]+K_{LA}^{AAB}[AAB]} [AAB]$ | $v_8 = k_{AAB} [AAB] [Ac]^2$ | $d_3 = b_{Ac} [Ac]$ |
| $v_{11} = \frac{\mu_{max}^{AAB_{Ac}} [Ac]}{[Ac]+K_{Ac}^{AAB}} [AAB]$ | | |

Microbial groups: yeast (Y), lactic acid bacteria (LAB) and acetic acid bacteria (AAB). Metabolites: glucose (Glc), fructose (Fru), ethanol (EtOH), lactic acid (LA) and acetic acid (Ac). Microbial groups and metabolites are expressed as concentrations, both within square brackets [ ]. Maximum specific growth rates $\mu_{max}^{i_n}$, correspond to the maximum growth rate of microbial group $i$, growing on substrate $n$. Substrate saturation constants $K_m^i$, correspond to the substrate saturation constant of microbial group $i$, growing on substrate $m$. Constant mortality rates $k_i$, correspond to mortality of microbial group $i$. Decay rates $d_j$, correspond to decay rate of metabolite $j$. All rates with the exception of $d_1$, $d_2$, $d_3$, $v_9$, $v_{10}$ and $v_{11}$, are part of the baseline model as proposed by Moreno-Zambrano *et al.* [9].

### 2.2.4. Mechanism 3: production of Ac by Y

Mechanism 3 (M3) is based on the evidence of among fermentation products that Y generate (e.g. ethanol, glycerol and carbon dioxide), Ac can be created through pyruvate metabolism and tricarboxylic acid cycle [3,4,16]. Besides, under controlled conditions, production of Ac by Y could explain concentrations of Ac that do not correspond to AAB's population sizes [17].

### 2.2.5. Mechanism 4: consumption of LA by Y

In mechanism 4 (MF4), during the first stage of fermentation, the Y population prevails due to the anaerobic conditions in the pulp. However, under an aerobic environment as during the third stage, yeasts such as *S. cerevisiae* are capable to oxidize LA to produce pyruvate [29,30]. Additionally, other species of yeast (e.g. *Pichia fermentans* and *Candida krusei*) can assimilate LA and produce EtOH.

### 2.2.6. Mechanism 5: over-oxidation of Ac by AAB

In mechanism 5 (M5), during the final stage of fermentation, AAB dominates microbial population by taking advantage of a fully aerobic environment, while consuming EtOH and LA previously produced by Y and LAB, respectively. Once EtOH has mostly diminished, it has been argued that AAB starts over-oxidizing Ac into carbon dioxide, which would lead to halting the cocoa fermentation process due to an increase of temperature that results in the declining of Y, LAB and AAB [3,4,6,31].

A comprehensive graphical representation of all proposed mechanisms is shown in figure 1*a*, full model including all mechanisms here proposed is shown in figure 1*b* and a detailed interpretation of all model parameters is shown in table 3.

## 2.3. Models iterations

For the purpose of checking the plausibility of different mechanisms working together, a series of model variants with combinations of M1, M2, M3, M4 and M5 were created, starting from the baseline model.

**Table 3.** Parameters of the cocoa bean fermentation baseline model and proposed mechanisms.

| parameter | mechanism | units | interpretation |
|---|---|---|---|
| $\mu_{max}^{Y_{Glc}}$ | B | $h^{-1}$ | maximum specific growth rate of Y on Glc |
| $\mu_{max}^{Y_{Fru}}$ | B | $h^{-1}$ | maximum specific growth rate of Y on Fru |
| $\mu_{max}^{Y_{LA}}$ | M4 | $h^{-1}$ | maximum specific growth rate of Y on LA |
| $\mu_{max}^{LAB_{Glc}}$ | B | $h^{-1}$ | maximum specific growth rate of LAB on Glc |
| $\mu_{max}^{LAB_{Fru}}$ | M2 | $h^{-1}$ | maximum specific growth rate of LAB on Fru |
| $\mu_{max}^{AAB_{EtOH}}$ | B | $h^{-1}$ | maximum specific growth rate of AAB on EtOH |
| $\mu_{max}^{AAB_{LA}}$ | B | $h^{-1}$ | maximum specific growth rate of AAB on LA |
| $\mu_{max}^{AAB_{Ac}}$ | M5 | $h^{-1}$ | maximum specific growth rate of AAB on Ac |
| $K_{Glc}^{Y}$ | B | $mg(Glc)g(pulp)^{-1}$ | substrate saturation constant of Y growth on Glc |
| $K_{Fru}^{Y}$ | B | $mg(Fru)g(pulp)^{-1}$ | substrate saturation constant of Y growth on Fru |
| $K_{LA}^{Y}$ | M4 | $mg(Fru)g(pulp)^{-1}$ | substrate saturation constant of Y growth on LA |
| $K_{Glc}^{LAB}$ | B | $mg(Glc)g(pulp)^{-1}$ | substrate saturation constant of LAB growth on Glc |
| $K_{Fru}^{LAB}$ | M2 | $mg(Fru)g(pulp)^{-1}$ | substrate saturation constant of LAB growth on Fru |
| $K_{EtOH}^{AAB}$ | B | $mg(EtOH)g(pulp)^{-1}$ | substrate saturation constant of AAB growth on EtOH |
| $K_{LA}^{AAB}$ | B | $mg(LA)g(pulp)^{-1}$ | substrate saturation constant of AAB growth on LA |
| $K_{Ac}^{AAB}$ | M5 | $mg(Ac)g(pulp)^{-1}$ | substrate saturation constant of AAB growth on Ac |
| $k_Y$ | B | $mg(EtOH)^{-1}h^{-1}$ | mortality rate constant of Y |
| $k_{LAB}$ | B | $mg(LA)^{-1}h^{-1}$ | mortality rate constant of LAB |
| $k_{AAB}$ | B | $mg(Ac)^{-2}h^{-1}$ | mortality rate constant of AAB |
| $Y_{Glc|Y}$ | B | $mg(Glc)mg(Y)^{-1}$ | Y-to-Glc yield coefficient |
| $Y_{Glc|LAB}$ | B | $mg(Glc)mg(LAB)^{-1}$ | LAB-to-Glc yield coefficient |
| $Y_{Fru|Y}$ | B | $mg(Fru)mg(Y)^{-1}$ | Y-to-Fru yield coefficient |
| $Y_{Fru|LAB}$ | M2 | $mg(Fru)mg(LAB)^{-1}$ | LAB-to-Fru yield coefficient |
| $Y_{EtOH|Y}^{Glc}$ | B | $mg(EtOH)mg(Y)^{-1}$ | Y-to-EtOH from Glc yield coefficient |
| $Y_{EtOH|Y}^{Fru}$ | B | $mg(EtOH)mg(Y)^{-1}$ | Y-to-EtOH from Fru yield coefficient |
| $Y_{EtOH|Y}^{LA}$ | M4 | $mg(EtOH)mg(Y)^{-1}$ | Y-to-EtOH from LA yield coefficient |
| $Y_{EtOH|LAB}^{Glc}$ | M2 | $mg(EtOH)mg(LAB)^{-1}$ | LAB-to-EtOH from Glc yield coefficient |
| $Y_{EtOH|LAB}^{Fru}$ | M2 | $mg(EtOH)mg(LAB)^{-1}$ | LAB-to-EtOH from Fru yield coefficient |
| $Y_{EtOH|AAB}$ | B | $mg(EtOH)mg(AAB)^{-1}$ | AAB-to-EtOH yield coefficient |
| $Y_{LA|LAB}^{Glc}$ | B | $mg(LA)mg(LAB)^{-1}$ | LAB-to-LA from Glc yield coefficient |
| $Y_{LA|LAB}^{Fru}$ | M2 | $mg(LA)mg(LAB)^{-1}$ | LAB-to-LA from Fru yield coefficient |
| $Y_{LA|AAB}$ | B | $mg(LA)mg(AAB)^{-1}$ | AAB-to-LA yield coefficient |
| $Y_{LA|Y}$ | M4 | $mg(LA)mg(Y)^{-1}$ | Y-to-LA yield coefficient |
| $Y_{Ac|LAB}^{Glc}$ | B | $mg(Ac)mg(LAB)^{-1}$ | LAB-to-Ac from Glc yield coefficient |
| $Y_{Ac|LAB}^{Fru}$ | M2 | $mg(Ac)mg(LAB)^{-1}$ | LAB-to-Ac from Fru yield coefficient |
| $Y_{Ac|AAB}^{EtOH}$ | B | $mg(Ac)mg(AAB)^{-1}$ | AAB-to-Ac from EtOH yield coefficient |
| $Y_{Ac|AAB}^{LA}$ | B | $mg(Ac)mg(AAB)^{-1}$ | AAB-to-Ac from LA yield coefficient |
| $Y_{Ac|Y}^{Glc}$ | M3 | $mg(Ac)mg(Y)^{-1}$ | Y-to-Ac from Glc yield coefficient |
| $Y_{Ac|Y}^{Fru}$ | M3 | $mg(Ac)mg(Y)^{-1}$ | Y-to-Ac from Fru yield coefficient |
| $Y_{Ac|AAB}$ | M5 | $mg(Ac)mg(AAB)^{-1}$ | AAB-to-Ac yield coefficient |
| $b_{EtOH}$ | M1 | $h^{-1}$ | decay rate of EtOH |
| $b_{LA}$ | M1 | $h^{-1}$ | decay rate of LA |
| $b_{Ac}$ | M1 | $h^{-1}$ | decay rate of Ac |

Microbial groups: yeast (Y), lactic acid bacteria (LAB) and acetic acid bacteria (AAB). Metabolites: glucose (Glc), fructose (Fru), ethanol (EtOH), lactic acid (LA) and acetic acid (Ac). B, M1, M2, M3, M4 and M5 refer to baseline model and mechanisms 1 to 5, respectively.

Hence, 31 MIs plus the baseline model were object of being fitted to experimental data under a Bayesian parameter estimation framework. Each MI is labelled according to the mechanisms involved. For example, the full model containing all five proposed regulatory schemes is labelled MI(1,2,3,4,5), while the baseline is labelled MI(0).

## 2.4. Kinetic parameter estimation

The number of parameters among MIs constructed over combination of mechanisms ranges from 24 in the baseline model, to 43 in the full model including all mechanisms. In each case, a general Bayesian framework was used to sample their posterior distributions, where means were taken as point estimates with their corresponding 95% credible intervals (CIs) [9].

### 2.4.1. Bayesian framework

First, let us consider any of our proposed deterministic MIs represented in a general form

$$\frac{\mathrm{d}x_i}{\mathrm{d}t} = f(x, \theta), \tag{2.9}$$

where $x$ represents a vector of state variables, $x_i$ is its $i$th component and the function $f(x, \theta)$ summarizes the dependence of the right-hand side of the ODEs on $x$ and all $k$ model parameters $[\theta_1, \theta_2, ..., \theta_k]$ contained in vector $\theta$.

If we assume that parameters $\theta$ are selected such that a set of data $\mathcal{Y}$ is described, a way to infer them is to compute the (posterior) probability of $\theta$ given $\mathcal{Y}$, $P(\theta \mid \mathcal{Y})$, which by applying Bayes' theorem, is equal to

$$P(\theta \mid \mathcal{Y}) = \frac{P(\mathcal{Y} \mid \theta) P(\theta)}{P(\mathcal{Y})}. \tag{2.10}$$

Here, since $P(\mathcal{Y})$ is a normalizing constant allowing the posterior density to integrate to one, equation (2.10) can be written in terms of the likelihood of observing $\mathcal{Y}$ given $\theta$, $P(\mathcal{Y} \mid \theta)$, and the prior distribution of vector $\theta$, $P(\theta)$, as

$$P(\theta \mid \mathcal{Y}) \propto P(\mathcal{Y} \mid \theta) P(\theta). \tag{2.11}$$

If we take into account that each component of $\mathcal{Y}$ contains $T$ time steps, with $N$ state variables being observed, equation (2.11) takes the form of a product over all series and each of their measured points as

$$P(\theta \mid \mathcal{Y}) \propto \prod_{i=1}^{N} \prod_{j=1}^{T} P(\mathcal{Y}_{i,j} \mid \theta) P(\theta). \tag{2.12}$$

Finally, as our purpose is to identify values of $\theta$ that lead to a best agreement between $\mathcal{Y}_{i,j}$ in equation (2.12) and $x_i(j)$ in equation (2.9), we can consider $\mathcal{Y}_{i,j}$ to be sampled from a normal distribution whose mean is equal to the model's prediction $f(x, \theta)$, with a standard deviation term, $\sigma$, (caused by noise of any kind) allowing us to reformulate the total posterior distribution as

$$P(\theta \mid \mathcal{Y}) \propto \prod_{i=1}^{N} \prod_{j=1}^{T} \mathcal{N}(f(x_{i,j}, \theta), \sigma) P(\theta). \tag{2.13}$$

Hence, by applying the total posterior distribution in equation (2.13), an extra parameter corresponding to a total standard deviation, $\sigma$, is also estimated.

### 2.4.2. Choice of priors

The regularization procedure of the data described in §2.1 permits to reduce the parameter search space in a convenient way for choosing the priors in ranges between 0 and approximately 1, which brings three main advantages: (i) independent prior distributions for each parameter can take the same form, (ii) by introducing scale information of the original units in which the parameters of the models are originally measured, we can formulate weakly informative priors capable of covering all possible values in the scaled space [9,32], and (iii) by avoiding diffuse priors, further model comparisons will be less likely to be affected by common problems, such as over-fitting [33] and ill-defined posteriors [34].

Hence, posterior distributions of $\theta$ and $\sigma$ were computed using a normal distribution with mean 0.5 and standard deviation of 0.3 for each element of the parameter vector $\theta$ and a Cauchy distribution with location 0 and scale of 1 for $\sigma$. With the purpose of avoiding estimates with negative values, both priors were truncated to the positive set of real numbers,

and
$$\left.\begin{array}{ll} \theta_k \sim \mathcal{N}(0.5,\ 0.3), & \theta_k > 0 \\ \sigma \sim \mathcal{C}(0,\ 1), & \sigma > 0. \end{array}\right\} \tag{2.14}$$

For a detailed description of prior distributions rescaled to the parameters' original units see electronic supplementary material, table S2.

### 2.4.3. Implementation

The fit of MIs to experimental data was performed with Stan [35] via RStan package in R [36,37]. Posterior distributions of $\theta$, $\sigma$ and $f(x_{i,j}, \theta)$, were obtained by Markov chain Monte Carlo (MCMC) no-U-turn sampler (NUTS) method [38]. Each model was treated as an initial value problem, where ODEs were solved by the built-in Stan numerical solver *rk45* for non-stiff systems by means of fourth- and fifth-order Runge–Kutta method [39,40] with relative and absolute tolerance values of $1 \times 10^{-6}$ for both, and a maximum number of steps of $1 \times 10^{4}$. All MIs were fitted to data by running four parallel Markov chains of 3000 iterations each, with 1000 of them used for warm-up. Sampling convergence was assessed by examining $\hat{R}$ statistic, bulk effective sample size (bulk-ESS) and tail effective sample size (tail-ESS) as described by Vehtari *et al.* [41]. In cases where either bulk-ESS or tail-ESS were rejected at first, calibration routine was rerun doubling iterations (2000 for warm-up, 6000 in total) before reporting non-convergence. Assessing autocorrelation of the sampled parameters was performed by means of averaging computed effective sample sizes over number of posterior draws ($\text{ESS}/\text{N}_{\text{draws}}$) and checking whether these were above 12.5%, meaning that as minimum 1000 ESS were obtained [42] (a more detailed explanation on the convergence criteria is described in electronic supplementary material, S4).

## 2.5. Model assessment

Under the umbrella of the proposed Bayesian framework, it is important to define what will be called from now on a *successful fit*. For this aspect, we consider as such, any MI fit that converged according to the criteria described in §2.4.3 and does not involve any reparametrization or use of distinct priors in cases were divergences of MCMC-NUTS or complete lack of sampling could arise as a result of complicated geometries imposed on the posterior distributions by inclusion of the assessed mechanisms over particular datasets. With this in mind, the quality of the models was assessed from two perspectives: (i) success of each MI across all data and (ii) predictive accuracy comparison of all MIs for each dataset.

On the one hand, Bayesian model averaging (BMA [43]) weights were computed using pseudo-BMA [44]. These weights, understood as representations of the relative probability of each MI [45], were then averaged over the total number of datasets that were fitted at least once by any MI, and used as a measure of the adequacy of each MI across all data. For computing the mean value of BMA weights ($\text{BMA}_w$), non-successful fits were assigned values of zero. Furthermore, an observed success rate (OSR) and expected success rate (ESR) were determined on the basis of times where the model was satisfactorily fit to a given dataset. OSR is then defined by the ratio of the number of successful fits and the total of datasets fitted by at least one MI. ESR, used to properly compare the success rates of models with only a single additional mechanism with those models containing a combination of mechanisms, is defined as the product of the OSRs of the elementary MIs. For model variant MI(1,2,5), for example, the expected success rate is then the product of the observed success rates of the elementary models MI(1), MI(2) and MI(5).

On the other hand, all MIs that were suitably fitted to each dataset were compared by means of Pareto-smoothed importance sampling leave-one-out cross validation (PSIS-LOO, [46]) with the aim of checking whether a certain MI could perform outstandingly better than its counterparts in terms of its predictive accuracy.

## 2.6. Principal components analysis

In an approach of linking our mathematical exploration of mechanisms with a real-life application (where principal component analysis (PCA) is a common approach), we studied whether projecting the obtained

vectors of kinetic parameters on the PCA space allows for a separation (and hence classification) of experiments according to fermentation features (e.g. cocoa beans' country of origin and used cultivar), which are of interest in both academic research and the chocolate industry.

Consequently, six main features of fermentation trials were taken into account as groups and analysed via PCA over parameter estimates for all successful MIs. Of these, three consist of multiple classes: (i) country of origin, (ii) cacao cultivar, and (iii) fermentation method. The other three features are binary: (iv) use of starter culture, (v) turning of fermenting mass, and (vi) controlled temperature during fermentation. Experiments with features reported as unknown or missing were not considered for any PCA. Only the posterior samples contained within their 95% credible interval (CI) from each chain in the MCMC-NUTS runs were taken into account to perform PCA. The assessment of groups within the PCA results was realized only for features with more than one successful fit.

Moreover, PCAs were also performed over subgroups of parameter defined by the type of parameter and its association with a certain microbial group. Considered subgroups consisted of: (i) all MI parameters, (ii) maximum specific growth rates, (iii) mortality rates, (iv) yield coefficients, (v) Y-related parameters, (vi) LAB-related parameters, and (vii) AAB-related parameters. No PCA was run over substrate saturation constants due to their known correlation with maximum growth rates [47]. All PCAs used mean-centred data with no scaling, given that solutions of MIs were determined over scaled time series.

Finally, pairwise squared Mahalanobis distances ($D_M$, [48]) were computed between grouping classes of each feature to quantify the magnitude of their separation. To achieve this, centroids of PCA scores from principal components 1 (PC1) and 2 (PC2) were computed for each $j$ grouping class within an $i$ feature and used to determine $D_M$ as

$$D_M(\mathrm{PC1}_{i,j}, \mathrm{PC2}_{i,j}) = (\bar{x}_1 - \bar{x}_2)^T \, \mathbf{S}^{-1} \, (\bar{x}_1 - \bar{x}_2), \tag{2.15}$$

where $\bar{x}_1$ and $\bar{x}_2$ are the centroid values of the scores of $\mathrm{PC1}_{i,j}$ and $\mathrm{PC2}_{i,j}$ respectively; and $\mathbf{S}^{-1}$ is the inverse of the covariance matrix between groups classes [48,49].

Both, PCA and $D_M$ were implemented in R using functions *prcomp* [36] and *pairwise.mahalanobis* [50], respectively.

# 3. Results

## 3.1. First assessment of the models

First, we want to understand how well the different models—the baseline model from [9] and the MIs containing one or more of the additional mechanisms—perform. In order to identify differences in the success rate of the model variants, we apply every MI to every fermentation dataset. Table 4 summarizes the result.

In general terms, MIs summed up to 1024 runs over 32 available datasets; of which, 207 resulted in successful fits with values of $\hat{R}$ below 1.05, bulk-ESS and tail-ESS higher than 100 indicating that convergence of the MCMC-NUTS was accomplished (see electronic supplementary material, tables S3–S5). Furthermore, in terms of autocorrelation, all successful fits showed averaged values of ESS/$N_{\mathrm{draws}}$ over 12.5% (see electronic supplementary material, table S6), indicating an acceptable ESS above 1000. A number of nine datasets reported by Lefeber *et al.* [51,52], Moreira *et al.* [18], Bastos *et al.* [53] and Racine *et al.* [54] were not possible to fit with any MI at all. The remaining 23 fermentation datasets constitute the scope of our further investigation (table 1). As an example, figure 2 shows one MI, MI(2,3) describing the time series of one of the datasets (*mywb3* from Papalexandratou *et al.* [19]).

A striking observation is that the vast majority of MIs involving M5 were not able to produce successful fits to experimental data. Among these, exceptions are datasets described by Papalexandratou *et al.* [14], *brwb1* and *brwb2*. Both were well fitted by MI(5), MI(1,5) and MI(1,2,3,5); while MI(2,5) and MI(3,5) fitted *brwb1* and MI(1,4,5) fitted *brwb2* only. Thus, MIs (4,5), (2,3,5), (2,4,5), (3,4,5), (1,2,4,5), (1,3,4,5), (2,3,4,5) and (1,2,3,4,5) could not describe any dataset at all (table 4).

## 3.2. Model success

Both approaches for assessing model success across all datasets, a formal one as BMA and our proposed measures OSR and ESR, showed similar conclusions over competing MIs.

**Table 4.** Summary of successful fits across 31 models iterations (MIs) and baseline.

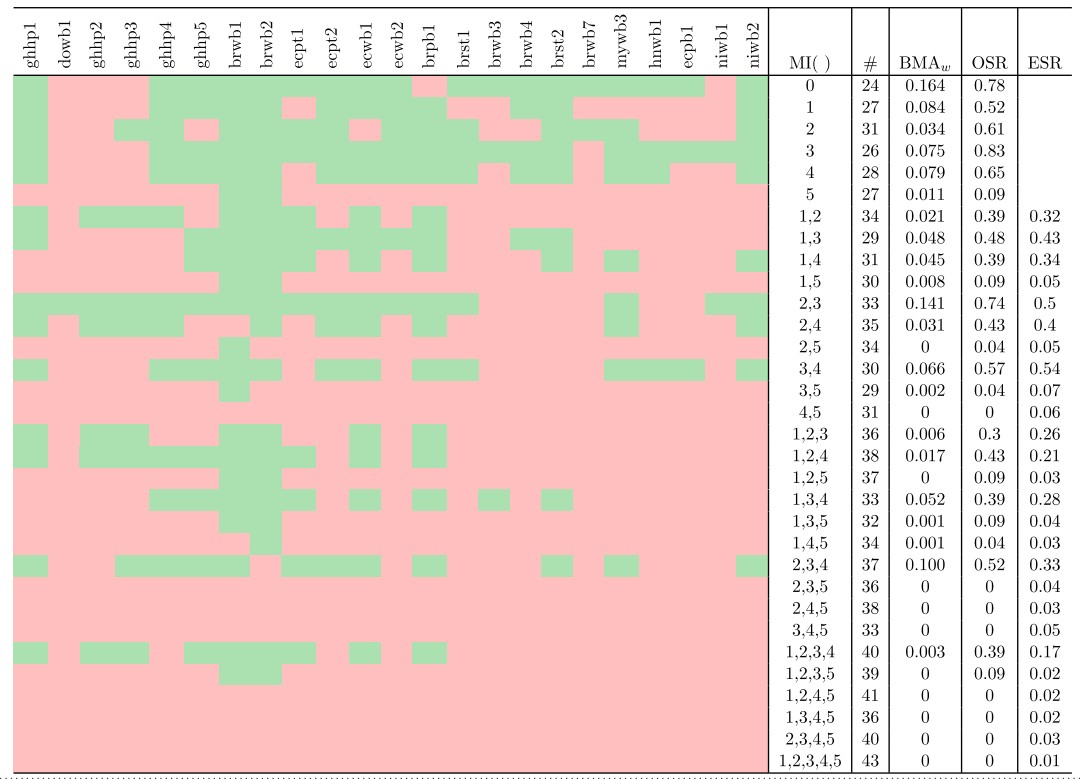

| MI( ) | # | BMA$_w$ | OSR | ESR |
|---|---|---|---|---|
| 0 | 24 | 0.164 | 0.78 | |
| 1 | 27 | 0.084 | 0.52 | |
| 2 | 31 | 0.034 | 0.61 | |
| 3 | 26 | 0.075 | 0.83 | |
| 4 | 28 | 0.079 | 0.65 | |
| 5 | 27 | 0.011 | 0.09 | |
| 1,2 | 34 | 0.021 | 0.39 | 0.32 |
| 1,3 | 29 | 0.048 | 0.48 | 0.43 |
| 1,4 | 31 | 0.045 | 0.39 | 0.34 |
| 1,5 | 30 | 0.008 | 0.09 | 0.05 |
| 2,3 | 33 | 0.141 | 0.74 | 0.5 |
| 2,4 | 35 | 0.031 | 0.43 | 0.4 |
| 2,5 | 34 | 0 | 0.04 | 0.05 |
| 3,4 | 30 | 0.066 | 0.57 | 0.54 |
| 3,5 | 29 | 0.002 | 0.04 | 0.07 |
| 4,5 | 31 | 0 | 0 | 0.06 |
| 1,2,3 | 36 | 0.006 | 0.3 | 0.26 |
| 1,2,4 | 38 | 0.017 | 0.43 | 0.21 |
| 1,2,5 | 37 | 0 | 0.09 | 0.03 |
| 1,3,4 | 33 | 0.052 | 0.39 | 0.28 |
| 1,3,5 | 32 | 0.001 | 0.09 | 0.04 |
| 1,4,5 | 34 | 0.001 | 0.04 | 0.03 |
| 2,3,4 | 37 | 0.100 | 0.52 | 0.33 |
| 2,3,5 | 36 | 0 | 0 | 0.04 |
| 2,4,5 | 38 | 0 | 0 | 0.03 |
| 3,4,5 | 33 | 0 | 0 | 0.05 |
| 1,2,3,4 | 40 | 0.003 | 0.39 | 0.17 |
| 1,2,3,5 | 39 | 0 | 0.09 | 0.02 |
| 1,2,4,5 | 41 | 0 | 0 | 0.02 |
| 1,3,4,5 | 36 | 0 | 0 | 0.02 |
| 2,3,4,5 | 40 | 0 | 0 | 0.03 |
| 1,2,3,4,5 | 43 | 0 | 0 | 0.01 |

Light green-coloured cells indicate successful fits. Light-red coloured cells indicate non-successful fits. Columns 'MI( )', '#', 'BMA$_w$', 'OSR' and 'ESR' refer to combination of mechanisms deployed in model iteration, number of parameters, averaged Bayesian model averaging weights, observed success rate and expected success rate, respectively.

From their computations, the baseline model (MI(0)) showed the highest values (BMA$_w$ = 0.164, OSR = 0.78). For MIs containing single mechanisms, M(1), M(2), M(3), M(4) and M(5), BMA$_w$ were 0.084, 0.034, 0.075, 0.079 and 0.011, while OSR values were 0.52, 0.61, 0.83, 0.65 and 0.09, respectively. Among more complex combinations of mechanisms, BMA$_w$ ranged between 0.00 to a maximum of 0.141 reached by the combination of M2 and M3 (MI(2,3)). Similarly, among these, MI(2,3) had a maximum OSR of 0.74 as listed in table 4.

Note that in general we expect a decrease of OSR with an increasing number of parameters in the model, due to higher complexity. Values for the single-mechanism MIs are therefore not directly comparable to the one of the baseline model. For even larger MIs (composite mechanisms), we have the ESR to partially correct for this.

Pertaining to ESRs, leaving out non-successful MIs, 2 out of 18 MIs showed higher values than their corresponding OSRs. These two MIs with higher ESRs correspond to iterations including M5 (i.e. MI(2,5) and MI(3,5)). Leaving aside MI(2,5) and MI(3,5) due to be the only exceptions of M5 ending up in successful fits, in overall combinations of mechanisms seems to lead to increases of their OSR over ESR on describing different datasets despite not overpassing the OSR of the baseline model (table 4).

## 3.3. Posterior predictions

Next, we resort to the distributions of posterior probabilities, in order to assess differences in the quality of the fit for the different MIs. Among the 23 datasets that were fitted by at least one MI, posterior predictions describe their dynamics remarkably well. In each data collection, despite the presence of highly influential observations and sampling rates ranging from 6 to 17 data points, time courses are simulated to an acceptable level. Again we refer to the example shown in figure 2, showing the fit of MI(2,3) to the dataset *mywb3* (see electronic supplementary material, figures S2–S17 for posterior predictions made by MI(2,3) and electronic supplementary material, figures S18–S23 for MIs, where M(2,3) was not suitable).

In terms of predictive accuracy among MIs fitted on each dataset, there were no outstanding differences on the basis of obtained PSIS-LOO deviance values that had overlapping standard errors

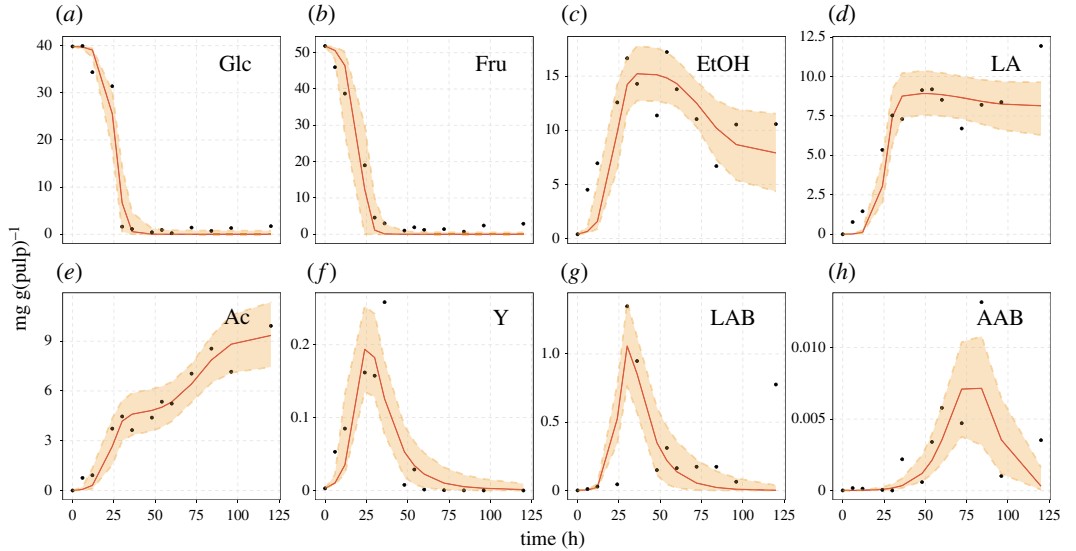

**Figure 2.** Posterior predictions of model iteration (MI) corresponding to mechanisms M2 and M3, MI(2,3), fitted to dataset *mywb3* reported by Papalexandratou *et al.* [19]. Metabolites: (*a*) glucose, (*b*) fructose, (*c*) ethanol, (*d*) lactic acid and (*e*) acetic acid. Microbial groups: (*f*) yeast, (*g*) lactic acid bacteria, and (*h*) acetic acid bacteria. Solid red lines represent posterior medians of the posterior predictions, solid black points denote experimental data and orange ribbon describe the 95% credible interval of posterior predictions.

between each other. Thus, is not surprising that for these cases posterior predictions resulted to be extremely similar (see electronic supplementary material, figure S24 for an example). Nevertheless, slight distinct PSIS-LOO were observed towards favouring MIs involving M1 in for datasets *brpb1*, *brwb4*, *brst2* and *niwb2* (see electronic supplementary material, figure S1). The latter is also evident by higher BMA weights by such MIs (see electronic supplementary material, table S7).

These subtle differences provided visually better fits by MI(1) with respect to MI(2,3) for datasets *brpb1* (see electronic supplementary material, figure S25). However, the same does not seem to be clear for *niwb2* where predictions made by M(1) and M(2,3) overlay each other with no evident improvements for either MI (see electronic supplementary material, figure S26).

## 3.4. Fermentation features

We now turn to the second question raised, namely whether the model parameters obtained by describing the datasets with all MIs are informative of the fermentation features behind the datasets. Via principal component analysis performed on the full parameter vectors or biologically meaningful subsets of the parameters, we want to assess whether distinct clusters emerge in agreement with differences in fermentation set-ups.

After dropping the use of a starter culture as a feature (see electronic supplementary material, S2), a total of 490 PCAs were performed from the remaining five features and seven parameters subsets. Note that MI(1,4) did not converge for datasets representing more than one used fermentation method, and MI(1), MI(1,2), MI(1,3), MI(1,4), MI(2,4), MI(1,2,3), MI(1,2,4), MI(1,3,4), MI(2,3,4) and MI(1,2,3,4) were not capable of describing datasets with more than one class of controlled temperature (see electronic supplementary material, figure S27).

In terms of group separation measured by $D_M$ for cases with more than one pairwise comparison, medians of all $D_M$ were computed ($\tilde{D}_M$) as a way to visualize the magnitude of separation as single values. From this analysis, it can be described in general terms that cultivar, temperature and fermentation method showed the highest $\tilde{D}_M$ values; while, origin countries and turning of fermenting mass showed almost no separation between groups (with the exception of a few cases). Details are provided in the following subsections (see also electronic supplementary material, figure S27).

### 3.4.1. Grouping of fermentation trials according to cultivar

PCAs with cultivar as feature of interest showed a consistent pattern of high values for $\tilde{D}_M$ with special emphasis on the subgroup of all parameters. With regard to MIs, clearer separations were the product of

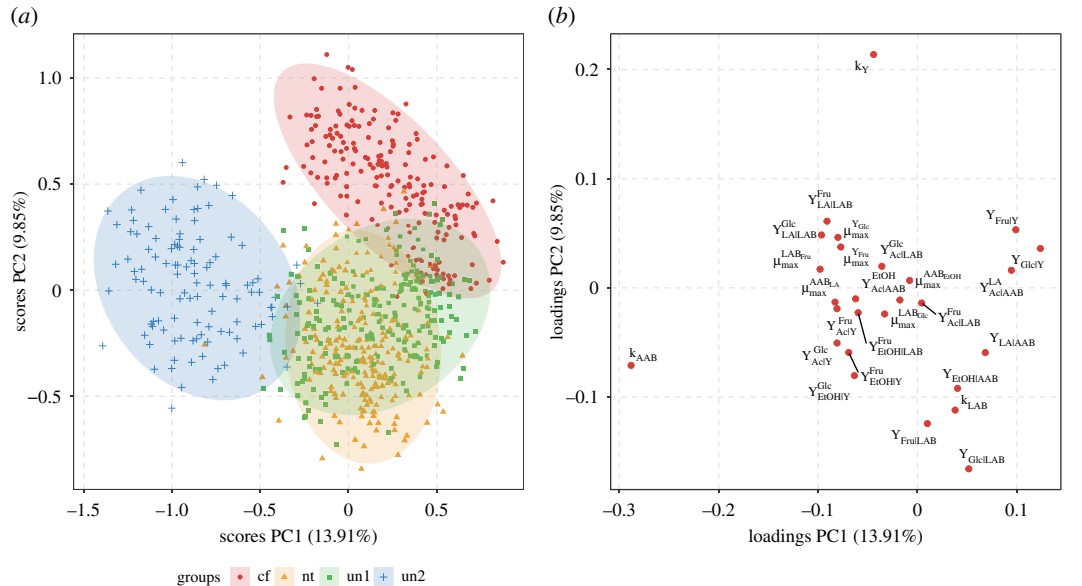

**Figure 3.** PCA score (*a*) and loading plot (*b*) from all parameters of model iteration MI(2,3), feature cultivar. For visualization purposes, scores of only 10% of posterior draws are shown. Criollo/Forastero (cf), Nacional/Trinitario (nt), unknown cultivar used by Camu *et al.* [13] (un1) and unknown cultivar used by Pereira *et al.* [16] (un2) are shown. Parameters located on the left and right with respect to 0 in PC1 loading plot determine differentiation between cf, nt and un2.

mostly complex MIs involving combinations of M2, M3 and M4 (see electronic supplementary material, figure S27, panel (*b*)).

Figure 3 shows a PCA plot for MI(2,3). Among the four cultivar varieties, three showed a clear separation, namely Criollo/Forastero, Nacional/Trinitario and *un2* with explained variances of 13.91% by the first PCA component (PC1) and 9.85% by the second component (PC2). From its loading plot (figure 3*b*), parameters with negative loadings in PC1, mainly $\mu_{max}^{Y_{Glc}}$, $\mu_{max}^{Y_{Fru}}$, $\mu_{max}^{LAB_{Glc}}$, $\mu_{max}^{LAB_{Fru}}$, $\mu_{max}^{AAB_{LA}}$, $\mu_{max}^{AAB_{EtOH}}$, $Y_{LA|LAB}^{Glc}$, $Y_{LA|LAB}^{Fru}$, $Y_{EtOH|LAB}^{Glc}$, $Y_{EtOH|LAB}^{Fru}$, $Y_{EtOH|Y}^{Glc}$, $Y_{EtOH|Y}^{Fru}$, $Y_{Ac|LAB}^{Glc}$, $Y_{Ac|AAB}^{EtOH}$, $Y_{Ac|Y}^{Glc}$, $Y_{Ac|Y}^{Fru}$, $k_Y$ and $k_{AAB}$ determine the classes separation.

### 3.4.2. Grouping of fermentation trials according to temperature control

Temperature control showed its highest $\tilde{D}_M$ values with MIs involving M2 (see electronic supplementary material, figure S27, panel (*e*)). The PCA of the set of all parameters for MI(2,3) resulted in a clear separation of groups determined by the use of controlled and non-controlled temperature with PC1 and PC2 scores explaining 11.87% and 8.76% of variance, respectively (figure 4). Likewise, the same set of parameters as in §3.4.1, showed negative loadings in PC1, indicating that these are defining the observed separation.

### 3.4.3. Grouping of fermentation trials according to method

Fermentation method as classification feature resulted in less clear separation patterns. Highest values of $\tilde{D}_M$ were observed for MI(2) (see electronic supplementary material, figure S27, panel (*c*)). In this case, a clear separation between fermentations carried out in stainless-steel tanks and the rest of fermentation methods can be seen with total explained variance of 16.61% by PC1 and 9.97% by PC2 (see electronic supplementary material, figure S28). Furthermore, parameters with positive loadings in PC1, namely $\mu_{max}^{Y_{Glc}}$, $\mu_{max}^{Y_{Fru}}$, $\mu_{max}^{LAB_{Glc}}$, $\mu_{max}^{LAB_{Fru}}$, $\mu_{max}^{AAB_{LA}}$, $\mu_{max}^{AAB_{EtOH}}$, $Y_{LA|LAB}^{Glc}$, $Y_{LA|LAB}^{Fru}$, $Y_{EtOH|Y}^{Glc}$, $Y_{EtOH|Y}^{Fru}$, $Y_{EtOH|LAB}^{Glc}$, $Y_{EtOH|LAB}^{Fru}$, $Y_{Ac|AAB}^{EtOH}$ and $k_{AAB}$ seem to influence the separation between stainless-steel tank and the rest of methods (see electronic supplementary material, figure S28, panel (*b*)).

### 3.4.4. Grouping of fermentation trials according to country of origin and turning of fermenting mass

In contrast to previous features, country of origin and turning of fermenting mass have lower $\tilde{D}_M$ values for single mechanisms and less cases of notoriously high distances. Instead, combinations of mechanisms and

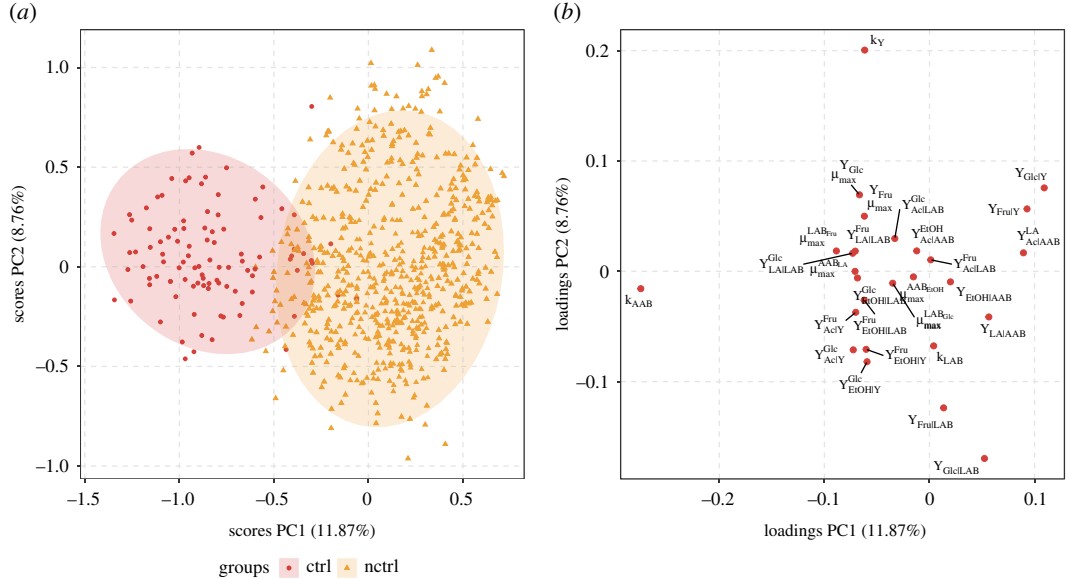

**Figure 4.** PCA score (*a*) and loading plot (*b*) from all parameters of model iteration MI(2,3), feature temperature. For visualization purposes, scores of only 10% of posterior draws are shown. Controlled temperature (ctrl) and non-controlled temperature (nctrl) are shown. Parameters located on the left and right with respect to 0 in PC1 loading plot determine differentiation between ctrl and nctrl.

other subgroups of parameters, rather than the whole set, led to better classes separations. For country of origin, PCAs over LAB-related parameters on MIs involving combinations of M1, M2 and M4 resulted in higher $\tilde{D}_M$. For turning of fermenting mass, there were no sizeable differences (see electronic supplementary material, figure S27, panels (*a*) and (*d*)). For this case PCA from MI(1,3,4) explains a total variance of 45.8% for PC1 and 25.89% for PC2. Furthermore, a clear separation between Brazil with respect to Ecuador and Ghana, is defined over PC1. From its loadings, parameters with positive loadings in PC1 ($\mu_{max}^{LAB|Glc}$, $Y_{LA|LAB}$ and $k_{LAB}$) are clearly denoting the separation between classes (see electronic supplementary material, figure S29). Lastly, turning of fermenting mass did not show any separation either for MI nor any subset of parameters.

# 4. Discussion

## 4.1. Model plausibility

We have assessed a series of mathematical model variants (or MIs) for cocoa bean fermentation in terms of two levels of plausibility: success of each run of a MI over a given dataset, and success of each MI to adequately describe the whole range of fermentation datasets. In this sense, failure in fitting a certain MI might be a consequence of practical non-identifiability of the parameters caused by weakly informative observations or priors, misspecification of the model [42], and even due to known limitations of MCMC-NUTS, such as poor chain mixing in presence of multimodal posteriors [55] and its higher sensitivity to model parametrization with respect to other samplers [56]. Success of a model, quantitatively represented as OSR per MI, highly depends on its capability to describe as many datasets as possible. Correspondingly, fitting failure becomes an indirect diagnostic tool that serve us to argue whether hypothesized regulatory interactions of the cocoa bean fermentation process are actually likely to be an influencing factor in the observed fermentation time series.

In general, the mechanisms discussed here have shown in most cases that their stand-alone and concomitant inclusion in the baseline model lead to convincing OSRs values, in particular for M2, M3 and M4. We can consider a lack of success in some runs involving these mechanisms is the product of numerically conflicting combinations of mechanisms rather than possible misspecification of the whole MI itself.

On the other hand, wide fitting failure patterns in runs involving M5 (table 4) and to a lesser extent also for M1, leads to the conclusion that these two mechanisms are not such significant influencing factors in the fermentation data studied here.

In the following, we will elaborate on the implications of fitting failure and plausibility of each mechanism considered in descending order relative to their OSR of the stand-alone inclusion in the baseline model.

Starting with M3, production of Ac by Y's metabolism has not been hypothesized from direct experimental measurements as is the case for other mechanisms. Instead, indirect kinetic studies of isolated strains have shown such an ability of some species of Y [16]. This property has also been argued to be a possible explanation of high Ac production yields where populations of AAB seemed incapable of producing such amounts, as proposed by Pereira *et al.* [17]. Hence, given the OSRs obtained from its inclusion in the series of MIs presented here, M3 can be considered as quantitative evidence backing up this role of Y during fermentation. We strongly believe that cases where these MIs did not succeed are the consequence of weakly informative priors incapable of being sampled properly.

With respect to M2, in light of the recent characterization of fructophilic lactic acid bacteria (FLAB) in cocoa bean fermentation processes [28], the level of OSR achievement of MIs involving M2 is not surprising. The existence of this bacterial group can then explain an apparent discrepancy in the amounts of Glc and Fru consumed during the process. Despite both substrates being depleted in parallel, Glc is usually consumed first as Y populations reach their end. Thus, uptake of remaining Fru might be a consequence of FLAB activity. In contrast to M3, fitting failure of M2 might be a result of weakly informative observations in some of the datasets with no time lag between Glc and Fru consumption.

Similarly to M3, the reasoning behind M4 relies on indirect characterization of Y strains capable of metabolizing LA into EtOH [12] via their known metabolic pathways producing pyruvate from LA [29,30] for further production of EtOH [4]. However, in contrast to M3, M4 obtained an appreciable OSR for its stand-alone iteration rather than for its occurrence jointly with either M2 and M3. As an example, consider MI(2,3) compared with MI(2,4) and MI(3,4). While MI(2,3) stands out as the MI with more than one mechanism with largest OSR (equal to 0.74), its counterparts involving M4 perform poorly with OSRs equal to 0.43 for MI(2,4) and 0.57 for MI(3,4). In our opinion, this counterintuitive performance of M4 when combined with M2 and M3 can be the result of numerical issues due to conflicting interactions of these mechanisms preventing the Bayesian optimization to sample from complicated geometries of the posterior target, rather than biological causes against M4.

MIs employing M1 and M5 resulted in the lowest OSRs among all, both stand-alone and combined with other mechanisms. However, important distinctions need to be made between these two. First, M1 is formulated on the basis of several experimental studies that have brought evidence of metabolites diffusing into the bean [11,13,14,16,17], e.g. EtOH, LA and Ac, besides evaporation and degradation processes not directly measured, but highly likely. In this sense, low success of MIs accounting M1 can be due to a lack of the models to describe dynamics of metabolites diffusing into the inner bean. In particular, the pure degradation mechanism included in our MI does not fully account for all possible sinks (e.g. due to diffusion and evaporation) of these substances. Cases in which M1 and its iterations resulted in successful fits, are those where clear decreases of EtOH, LA and Ac are visible in the time series (see electronic supplementary material, figures S18, S19 and S25).

Second, regardless of the widely accepted mechanism of AAB consuming Ac once EtOH concentration has reached minimum levels in the fermenting mass [3,4,6,31], we have found solid quantitative evidence through assessing M5 that such a phenomenon has a very small impact on the process dynamics and thus is unlikely. By the end of fermentation, the AAB populations have been highly diminished and, in most of the datasets considered here, drops in Ac concentration were seldom reported. In other words, if M5 actually has an impact on the whole process, it would be necessary that AAB counts remain viable up to its completion in order to deplete Ac. This observation can also explain the few exceptions, in which M5 led to successful fits. In total, the vast majority of datasets showed minimal counts of AAB even before the penultimate day of fermentation, with the exception of *brwb1* and *brwb2* reported by Papalexandratou *et al.* [14], where drops of AAB counts are quite abrupt by its last day, limiting the capability of these MIs to simulate a complete diminution of its population (see electronic supplementary material, figures S8 and S9).

Finally, after reviewing all above-mentioned causes of fitting failure and how they could have affected success ratios of each MI, it is plausible to assume that harsher combinations of such causes are responsible for failing to fit any model to nine datasets (see §3.1) as well for some others that were scarcely described, e.g. *dowb1* and *niwb2*.

## 4.2. Interpretation of the posterior distributions of model parameters

Here, we would like to discuss general aspects of the posterior distributions and how they allow us to further assess the different MIs. We will focus on the agreement of the obtained posteriors with values in the literature as well as practical non-identifiability of parameters.

For parameter agreement, let us consider the set of posterior distributions for MI(2,3) fitted over several datasets (see electronic supplementary material, table S8). As stated in [9], among all estimated posteriors, few did not agree with reported estimates in the literature. Those which culminated in values far away from reported ranges (more than 10-fold) as well as others, whose biological plausibility is unlikely, suggest evidence of practical non-identifiability, as evidenced by posteriors with wide credible intervals. This might be due to weakly informative observations that do not capture entirely the dynamics of the included mechanisms [9,42].

This assumption seems to be supported by parameters' posteriors under the scope of different MIs. By the inclusion of extra terms acting upon the dynamics, in which non-identifiable parameters are suspected to exist, their wide ranges should be visibly reduced. In fact, focusing on examples already mentioned in [9] (particularly $Y_{LA|AAB}$, $Y_{EtOH|AAB}$ and $Y_{Ac|AAB}^{LA}$), we can see this reduction across different MIs describing datasets *ghhp1*, *ecwb1*, *ecwb2* and *brpb1* (see electronic supplementary material, figure S30).

Nevertheless, inclusion of extra parameters does not entirely eliminate non-identifiability, which suggests the need for more informative priors for further developments of cocoa bean fermentation modelling.

## 4.3. Grouping of fermentation features

We have seen that fermentation features can be distinguishable with respect to all parameter estimates derived from their posterior distributions, especially those from MI(2) and MI(2,3). From our perspective, this is a clear illustration, how ODE-based modelling, rather than the usual methods of chemical fingerprinting [57–59], can demonstrate differences in features of the process.

An example is the clear differences in fermentation features identified through kinetic parameter estimates of MI(2,3) for cultivar, controlled temperature and fermentation methodology. Regarding cultivar, similar findings have been reported for biochemical characterization studies where the same cultivars used in different countries have shown to be part of similar classes within PCA [57]. This would then explain why country of origin used as a feature, did not result in clear separation patterns. Instead, only few subsets of parameters for certain MIs ended up in clear group separations in that case, as it happens for the LAB-related parameters in MI(1,3,4) (see §3.4.4). This could be an indicator that indeed dynamics of different LAB populations are linked to the location where fermentation took place [5].

In a similar fashion, grouping of fermentation trials dependent on whether they were performed under controlled temperature settings, reflects how kinetic parameters of MI(2,3) might be influenced by this feature. In this regard, explicit inclusion of temperature in these models would be a natural option. From our point of view, we firmly consider that incorporation of temperature in this modelling scheme would be beneficial in case remarkable improvements on the assessing statistics presented here were seen. However, explicitly incorporating temperature as a dynamical variable did not dramatically change either PSIS-LOO, posterior predictions or parameter ranges towards more biological plausibility (see electronic supplementary material, S5).

Furthermore, classification of trials with respect to methodology also tends to lead to a clear separation with a special emphasis on trials performed in stainless-steel tanks fitted with MI(2). This finding suggests that the use of stainless-steel tanks affects kinetic parameters, making them distinct from other methods. Besides, it becomes an indication that inclusion of M2 seems to be driving this difference, as well as other feature discriminations. The latter observation is based on the loading plots that for all these PCAs are determined by parameters related to M2, such as $\mu_{max}^{LAB_{Fru}}$, $Y_{EtOH|LAB}^{Fru}$ and $Y_{LA|LAB}^{Fru}$.

Lastly, it is worth remarking that including all parameters' posterior draws contained in their 95% CI for conducting PCAs plays an important role in finding more meaningful differences between the features. This conclusion comes from a former approach where we performed PCAs over the medians of each chain of successful MIs instead. In such an exercise, besides obvious quantitative differences between its PCs and their counterparts reported here (see electronic supplementary material, table S9), the same classification of the features was observed, with the exception of a noticeable separation between all countries (see electronic supplementary material, figure S31) rather than the overlap between Ghana and Ecuador shown in electronic supplementary material, figure S29, which could have been mistakenly ignored if it were not by considering the posterior distributions' uncertainties.

## 5. Conclusion

The series of MIs presented here constitute a first kinetic exploration of the plausibility of regulatory dynamics of cocoa bean fermentation not considered in our previous modelling [9], but long reported

and hypothesized in the literature. Thus, it allows us to evaluate the plausibility of various mechanisms in a stand-alone and concomitant manner. Among the five mechanisms discussed here, M2 (consumption of fructose by lactic acid bacteria) and M3 (production of acetic acid by yeast) have gathered the strongest support in our investigation.

Our scheme also allows us to conclude that loss of metabolites by physical phenomena (M1) is quite minimal, relative to their consumption and formation rates emphasizing the importance of microbial biochemical processes. Furthermore, it also offers quantitative evidence that a widely hypothesized mechanism, M5 (over-oxidation of acetic acid by acetic acid bacteria), does not agree with experimental data.

With reference to fermentation features, the rich set of parameter estimation results grants for interpretation on three levels: (i) We find that the parametrized time courses separate different fermentation features with different quality. Across all models, origin countries seem to only have a small influence on systematic time course differences. By contrast, temperature and cultivar seem to have a strong effect on fermentation dynamics (and hence on systematic differences in the resulting parameter vectors). (ii) Orthogonal to this view, we can assess which model versions lead to better discrimination fermentation features compared with the basic fermentation model from [9]. This complements our assessment of parameter estimation success, which is summarized in table 4. (iii) By splitting parameters into groups, we can assess the involvement of certain microorganisms in the systematic differences between fermentation features.

Lastly, this work commends that in a pure sense of describing fermentation dynamics in the pulp, inclusion of temperature as a dynamical variable does not add improvements to fits obtained under the proposed scheme. However, future advancements in cocoa bean fermentation modelling might find it necessary to take it into account for a more refined description of more detailed experimental data and to capture its reported importance in mediating important dynamics, for instance, its role in diffusion processes of acids into the bean [60].

Data accessibility. Data and relevant code for this research work are stored in GitHub https://github.com/mmorenozam/ecbfm and have been archived within the Zenodo Repository https://zenodo.org/record/5510370 [61]. Supplementary results are provided in electronic supplementary material [62].

Authors' contributions. M.M.-Z.: conceptualization, data curation, formal analysis, investigation, methodology, software, validation, visualization, writing—original draft, writing—review and editing; M.S.U.: funding acquisition, project administration, supervision, validation, writing—original draft, writing—review and editing; M.-T.H.: conceptualization, funding acquisition, methodology, project administration, resources, supervision, validation, writing—original draft, writing—review and editing. All authors gave final approval for publication and agreed to be held accountable for the work performed therein.

Competing interests. We declare we have no competing interests.

Funding. This work was funded by Barry Callebaut through the Cocoa Metabolomics (COMETA) project at Jacobs University Bremen.

Acknowledgements. We gratefully acknowledge Dr Ben Bales, Dr Bob Carpenter and the rest of the Stan Development team for their advice and fruitful discussions in the Stan Forums (https://discourse.mc-stan.org/). M.M.-Z. acknowledges Dr Silvana Beltrán for her support through the conduct of this research.

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
