## [Peer Review File · Royal Society Open Science]

Review History

RSOS-210274.R0 (Original submission)

Review form: Reviewer 1

Is the manuscript scientifically sound in its present form?

Yes

Are the interpretations and conclusions justified by the results?

Yes

Is the language acceptable?

Yes

Do you have any ethical concerns with this paper?

No

Have you any concerns about statistical analyses in this paper?

Yes

Recommendation?

Major revision is needed (please make suggestions in comments)

Comments to the Author(s)

Please see the attached Report.pdf (Appendix A).

Decision letter (RSOS-210274.R0)

Dear Mr Moreno-Zambrano

The Editors assigned to your paper RSOS-210274 "Exploring cocoa bean fermentation mechanisms by kinetic modelling" have now received comments from reviewers and would like you to revise the paper in accordance with the reviewer comments and any comments from the Editors. Please note this decision does not guarantee eventual acceptance.

Please submit your revised manuscript and required files (see below) no later than 21 days from today's (ie 24-Aug-2021) date. Note: the ScholarOne system will 'lock' if submission of the revision is attempted 21 or more days after the deadline. If you do not think you will be able to meet this deadline please contact the editorial office immediately.

on behalf of Professor Len Thomas (Associate Editor) and Mark Chaplain (Subject Editor)
openscience@royalsociety.org

Associate Editor Comments to Author:

Comments to the Author:

Thank you for submitting your paper to Royal Society Open Science.

We've received one referee report on your paper. Please ensure that you address their concerns within a point-by-point response to be included upon re-submission. Please also ensure that you include a version of your revised paper which contains either tracked or highlighted changes you've made upon revision.

Reviewer comments to Author:

Reviewer: 1

Comments to the Author(s)

Please see the attached Report.pdf

===PREPARING YOUR MANUSCRIPT===

===PREPARING YOUR REVISION IN SCHOLARONE===

Author's Response to Decision Letter for (RSOS-210274.R0)

See Appendix B.

RSOS-210274.R1 (Revision)

Review form: Reviewer 1

Is the manuscript scientifically sound in its present form?

Yes

Are the interpretations and conclusions justified by the results?

Yes

Is the language acceptable?

Yes

Do you have any ethical concerns with this paper?

No

Have you any concerns about statistical analyses in this paper?

No

Recommendation?

Accept with minor revision (please list in comments)

Comments to the Author(s)

This revised version of the paper addresses correctly the concerns of the original review. In my opinion, the paper is ready for publication. I have a few additional minor comments:

- There is still a problem with considering that "few thousand iterations" is sufficient for "convergence". Note that even with independent MC sampling, one needs to take into consideration the variance of the statistic being approximated. The use of magic numbers like 1,000 or 100,000 samples is widespread but is simply wrong.
- I'm ok with the authors a practical approach to defining successful fit for which the MCMC-NUTS did not fail. With the new wording, this seems reasonable. However, please clarify that this is due to the limitations of the MCMC-NUTS: The authors of Stan would like to make us believe that if their sampler cannot handle a posterior distribution, then the user's posterior is wrong and the user must change their priors, models etc. This is a ridiculous and very convenient argument by the authors of Stan to justify their questionable sampler (which the authors of this paper are all very keen to defend).
- It will be very illustrating to now mention in the paper that, regarding the PCA analysis, inclusion of the uncertainty in posterior, using the MCMC draws, does make a difference in the analysis.

Decision letter (RSOS-210274.R1)

Dear Mr Moreno-Zambrano

On behalf of the Editors, we are pleased to inform you that your Manuscript RSOS-210274.R1 "Exploring cocoa bean fermentation mechanisms by kinetic modelling" has been accepted for publication in Royal Society Open Science subject to minor revision in accordance with the referees' reports. Please find the referees' comments along with any feedback from the Editors below my signature.

Please submit your revised manuscript and required files (see below) no later than Friday 14 January. Note: the ScholarOne system will 'lock' if submission of the revision is attempted after the deadline. If you do not think you will be able to meet this deadline please contact the editorial office immediately.

on behalf of Professor Len Thomas (Associate Editor) and Mark Chaplain (Subject Editor)
openscience@royalsociety.org

Reviewer comments to Author:

Reviewer: 1

Comments to the Author(s)

This revised version of the paper addresses correctly the concerns of the original review. In my opinion, the paper is ready for publication. I have a few additional minor comments:

- There is still a problem with considering that "few thousand iterations" is sufficient for "convergence". Note that even with independent MC sampling, one needs to take into consideration the variance of the statistic being approximated. The use of magic numbers like 1,000 or 100,000 samples is widespread but is simply wrong.

- I'm ok with the authors a practical approach to defining successful fit for which the MCMC-NUTS did not fail. With the new wording, this seems reasonable. However, please clarify that this is due to the limitations of the MCMC-NUTS: The authors of Stan would like to make us believe that if their sampler cannot handle a posterior distribution, then the user's posterior is wrong and the user must change their priors, models etc. This is a ridiculous and very

convenient argument by the authors of Stan to justify their questionable sampler (which the authors of this paper are all very keen to defend).

- It will be very illustrating to now mention in the paper that, regarding the PCA analysis, inclusion of the uncertainty in posterior, using the MCMC draws, does make a difference in the analysis.

===PREPARING YOUR MANUSCRIPT===

one version should clearly identify all the changes that have been made (for instance, in coloured highlight, in bold text, or tracked changes);

===PREPARING YOUR REVISION IN SCHOLARONE===

-- If you are requesting an article processing charge waiver, you must select the relevant waiver option (if requesting a discretionary waiver, the form should have been uploaded, see 'File upload' above).

-- If you have uploaded any electronic supplementary (ESM) files, please ensure you follow the guidance at <https://royalsociety.org/journals/authors/author-guidelines/#supplementary-material> to include a suitable title and informative caption. An example of appropriate titling and captioning may be found at https://figshare.com/articles/Table_S2_from_Is_there_a_trade-off_between_peak_performance_and_performance_breadth_across_temperatures_for_aerobic_scope_in_teleost_fishes_/3843624.

Author's Response to Decision Letter for (RSOS-210274.R1)

See Appendix C.

Decision letter (RSOS-210274.R2)

Dear Mr Moreno-Zambrano,

I am pleased to inform you that your manuscript entitled "Exploring cocoa bean fermentation mechanisms by kinetic modelling" is now accepted for publication in Royal Society Open Science.

on behalf of Professor Len Thomas (Associate Editor) and Mark Chaplain (Subject Editor)
openscience@royalsociety.org

Appendix A

Referee review: **Exploring cocoa bean fermentation mechanisms by kinetic modelling** by Mauricio Moreno-Zambrano, Matthias S. Ullrich and Marc-Thorsten Hutt. Submitted to the consideration of *Royal Society Open Science*.

The authors present a comparative study of a base line ODE model for cocoa fermentation with several other models varying in complexity and qualitative differences. The added complexities in the mechanistic model are nicely summarized in Figure 1, with 5 added mechanisms into the fermentation process signifying adding terms to the rhs of the system of ODE's. The 5 added variants are considered to create alternative models, leading to 31 possible alternatives along with the base line model.

1. L. 78-83: Note that proper data modeling and careful consideration of parameter units are common strategies for obtaining 'comparable' estimates from different data sets. Data pre-processing may well be sound, but the authors need to explain in more detail why their models do not report inherent 'comparable' parameters and why this preprocessing is preferred and correct. Can the models be re-parametrized to have comparable parameters?
2. Diffuse priors may become a big problem in model comparison; extensive literature has been written by the Bayesian community on this respect. Please explain prior selection in more detail.
3. Why RK45 is used. What time step size was used? This is a very old method, far more advanced methods are available with implicit and automatic time steps.
4. L. 189: Why the burn in or warm up was left at 1,000 iterations? Why such a small sample size? (4 parallel chains of 3,000, why use parallel chains?). Note that this would be quite short for an independent sampler, let alone with the correlation inherent in the MCMC. What is then the effective sample size?
5. Note that the corner stone of the paper is the model fit- model assessment. Section 2.5 is only very briefly explained and unclear. The authors should explain in far more detail their approach. For example, what is a "successful fit" (l. 199)?

Note that there is a formal Bayesian approach to model adequacy-model selection: The posterior probability of each model is proportional to its normalization constant, see <https://doi.org/10.1214/>

ss/1009212519 for example. All other approaches are approximations to the latter (e.g. the BIC, DIC etc.). The problem is estimating such normalizations constants using MCMC, which remains a difficult problem, still, see <https://doi.org/10.1198/106186008X320258> . The authors should try to use this formal approach, before resorting to other more ad hoc or heuristic methods.

6. Please explain in more detail section 2.6. Why the PCA is done? What is its purpose? It uses all data sets and all models also or only the winner model? Can the authors use the uncertainty in the posterior and not only use the medians (a point estimate)?
7. The authors seem to trust nearly blindly the statistics \hat{R} etc. **The authors are wrong here in believing that a bad convergence in the MCMC means a bad model.** The model can be very good but results in a difficult posterior to sample from and the MCMC method itself is what is inadequate. The U-turn sampler is not bullet proof, nor any of the other methods programmed in Stan. Difficult posteriors (multimodal, too correlated) are difficult to sample for any method and the methods in Stan are no exceptions (btw contrary to what the authors of Stan seem to claim). This misconception is also apparent in the Discussion, l. 354.

On the contrary, the posterior may be very simple (nice unimodal bell shape) to sample from, the MCMC be excellent but the model itself be useless.

Conceptually, the authors need to separate MCMC convergence and model fit. The former needs to be assured first, if no convergence is attained, a larger sample size may be needed or a different sampler altogether. There are other multipurpose samplers that may be used, like DRAM or the t-walk MCMC samplers. Once MCMC convergence has been attained then model adequacy may be analyzed.

I am confident that the authors results will hold, in general, since a detailed process of model assessment was done, but a more methodologically and conceptually sound analysis is in order.

Appendix B

Response to Reviewer's Comments on the Revision of the Manuscript

Exploring cocoa bean fermentation mechanisms by kinetic modelling

Mauricio Moreno-Zambrano¹, Matthias S. Ullrich¹, and Marc-Thorsten Hütt¹

¹Department of Life Sciences and Chemistry, Jacobs University Bremen, Campus Ring 1, 28759 Bremen, Germany

Dear Editors,

We thank the reviewer for their comments and recommendations. All the reviewer's concerns were carefully considered, leading to an updated version of our manuscript. In this letter, you will find a detailed response to each of the observations of the reviewer. In each case, we have mentioned where and how the updated version of the manuscript meets reviewer's recommendations.

The original reviews are reproduced in black font, while our response is given in blue font.

Mauricio Moreno-Zambrano (for the authors)

Reviewer 1:

The authors present a comparative study of a base line ODE model for cocoa fermentation with several other models varying in complexity and qualitative differences. The added complexities in the mechanistic model are nicely summarized in Figure 1, with 5 added mechanisms into the fermentation process signifying adding terms to the rhs of the system of ODE's. The 5 added variants are considered to create alternative models, leading to 31 possible alternatives along with the base line model.

1. L. 78-83: Note that proper data modeling and careful consideration of parameter units are common strategies for obtaining 'comparable' estimates from different data sets. Data pre-processing may well be sound, but the authors need to explain in more detail why their models do not report inherent 'comparable' parameters and why this preprocessing is preferred and correct.

We agree with the reviewer that our data-processing steps need more clarification. Answering part by part their concerns, we would like to mention that we indeed reported comparable parameters with those available for single-strained microbial growth studies in literature. Specifically, we made use of a re-scaled version of the obtained parameters of model iteration (MI) involving mechanisms 2 and 3 (MI(2,3)) in Section 4.2 and Supplementary Table S6 (now Supplementary Table S8, after reviewer's comments) when discussing parameter agreement between those obtained from our Bayesian framework and their corresponding values for single-strained microbial experiments. Thus, in order to give the reader a better explanation, we have changed paragraph mentioned by the reviewer as follows:

In all cases, population growth of Y, LAB and AAB were transformed from log base 10 of colony forming units ($\log_{10}(\text{CFU})$) to milligrams of microbial group (MG) per gram of pulp ($\text{mg}(\text{MG})\text{g}(\text{pulp})^{-1}$) as these are the units in which most of kinetic single-strained microbial growth studies report their dynamics as well as their dependent constants, i.e., specific maximum growth and mortality rates, and yield coefficients. Moreover, with the purpose of facilitating the estimation of models' parameters by avoiding numerical issues during their calibration, all time series were scaled by dividing each observation by its own maximum value. Hence, obtained parameter estimates were re-scaled to their original units by using simple transformations for their further comparison with previously reported values for single-strained microbial studies (see Supplementary Material, Section 1) [1].

Furthermore, we have added in the Supplementary material an extra section (now as Section 1 of this material) where we illustrate a simple example of how the re-scaling of the obtained parameters is performed to their original units as follows:

1 Parameter estimates re-scaling to their original units

Consider the kinetics of a given x variable in any of our proposed models, its concentration $[x]$ at any time and its maximum concentration $[x]_{max}$ within the whole interval in which the fermentation took place.

We can then let x' represent a transformation of x as

$$x' = \frac{[x]}{[x]_{max}}, \quad (1)$$

then, the time derivative of x' will be given by

$$\frac{dx'}{dt} = \frac{1}{[x]_{max}} \frac{d[x]}{dt}, \quad (2)$$

where $[x]_{max}$ is the maximum concentration of any of the state variables depending on which ODE of a proposed model is expressed in the form of Eq. (2).

Generalizing, one can express all ODEs of a model in the form of Eq. (2). As an example, let's express Eq. (1.MM) for the time derivative of glucose (Glc) as a scaled version in accordance to Eq. (2)

$$\frac{1}{[Glc]_{max}} \frac{d[Glc]}{dt} = -Y_1 \frac{\mu_1 \frac{[Glc]}{[Glc]_{max}}}{\frac{[Glc]}{[Glc]_{max}} + K_{s1}} \frac{[Y]}{[Y]_{max}} - Y_2 \frac{\mu_3 \frac{[Glc]}{[Glc]_{max}}}{\frac{[Glc]}{[Glc]_{max}} + K_{s3}} \frac{[LAB]}{[LAB]_{max}}, \quad (3)$$

where $\mu_1, \mu_3, K_{s1}, K_{s3}, Y_1$ and Y_2 are the solutions of this ODE parameters upon the scaled time series of Glc (according to Eq. (1)) for specific maximum growth rates (μ_i), substrate saturation constants (K_{s_i}) and yield coefficients (Y_i) respectively.

In other words, all aforementioned terms are scaled posterior distributions resulting from our Bayesian framework for which re-scaled versions to their original units can be accomplished by properly working out the scaled ODE.

Hence, Eq. 3 can be simplified to

$$\frac{d[Glc]}{dt} = -Y_1 \frac{[Glc]_{max}}{[Y]_{max}} \frac{\mu_1 [Glc]}{[Glc] + K_{s1} [Glc]_{max}} [Y] - Y_2 \frac{[Glc]_{max}}{[LAB]_{max}} \frac{\mu_3 [Glc]}{[Glc] + K_{s3} [Glc]_{max}} [LAB] \quad (4)$$

where the terms $\mu_1, \mu_3, K_{s1} [Glc]_{max}, K_{s3} [Glc]_{max}, Y_1 \frac{[Glc]_{max}}{[Y]_{max}}$ and $Y_2 \frac{[Glc]_{max}}{[LAB]_{max}}$ are equivalent to $\mu_{max}^{Y_{Glc}}, \mu_{max}^{LAB_{Glc}}, K_{Glc}^Y, K_{Glc}^{LAB}, Y_{Glc|Y}$ and $Y_{Glc|LAB}$ respectively and the left hand side of the equation corresponds to the state variable with no scaling.

Similarly, transformation factors as those previously described can be then obtained for the rest of parameters in any given MI presented in this study.

Can the models be re-parametrized to have comparable parameters?

Scaling of the data obtained from each of the fermentation trials represents itself a re-parametrization of the models in the way we have set up the Bayesian framework. In this sense, due to the differences between orders of magnitude of time series reported for microbial growth and metabolites (e.g., the maximum concentrations of acetic acid bacteria and ethanol by Camu et al. [2] are $0.0019 \text{ mg g(pulp)}^{-1}$ and $22.4920 \text{ mg g(pulp)}^{-1}$ respectively [1]), data scaling offers the advantage of setting a more general scheme of choosing priors within a Bayesian framework. In this way we do not require other model re-parametrizations that could end up with an increasing number of parameters (due to individual priors for each variable). For example, a usual practice of a simpler re-parametrization for parameters describing the dynamics of state variables, which highly differ between each other, can be done by multiplying microbial-related parameters (e.g., yield coefficients and substrate saturation constants) by arbitrary values that could lesser the effect of the magnitude in which metabolites differ from the measurements of microbial growth. However, this approach becomes difficult to implement when fitting several datasets, where those arbitrary values need to be set one by one depending on how much do the microbial time series differ with respect to their corresponding metabolite kinetics across different trials leading also to more complicated prior values choices. Finally, even if such an approach would succeed, will also require a transformation to the parameter real units for the sake of comparison. The latter, as we have above described, is also possible with our re-parametrization choice.

2. Diffuse priors may become a big problem in model comparison; extensive literature has been written by the Bayesian community on this respect. Please explain prior selection in more detail.

We agree with the reviewer that using diffuse priors may become a problem in model comparison. However, the limited knowledge regarding the kinetics of the cocoa bean fermentation process impedes the use of more informative priors. Besides, we consider that prior distributions in a regularized parameter space as the depicted in Section 2.4.2 (thanks to the scaling of the original data) have a weakly informative nature rather than a diffuse one. Considering this, and following the reviewer's concern, we have rephrased Section 2.4.2 as follows.

2.4.2 Choice of priors

The regularization procedure of the data described in Section 2.1 permits to reduce the parameter search space in a convenient way for choosing the priors in ranges between 0 to ≈ 1 , which brings three main advantages: (1) independent prior distributions for each parameter can take the same form, (2) by introducing scale information of the original units in which the parameters of the models are originally measured, we can formulate weakly informative priors capable of covering all possible values in the scaled space [1, 3], and (3) by avoiding diffuse priors, further model comparisons will be less likely of being affected by common problems, such as over-fitting [4] and ill-defined posteriors [5].

Hence, posterior distributions of θ and σ were computed using a normal distribution with mean 0.5 and standard deviation of 0.3 for each element of the parameter vector θ and a Cauchy distribution with location 0 and scale of 1 for σ . With the purpose of avoiding estimates with negative values, both priors were truncated to the positive set of real numbers,

$$\begin{aligned}\theta_k &\sim \mathcal{N}(0.5, 0.3), & \theta_k > 0 \\ \sigma &\sim \mathcal{C}(0, 1), & \sigma > 0.\end{aligned}\tag{5}$$

For a detailed description of prior distributions re-scaled to the parameters' original units see Supplementary Table S2.

3. Why RK45 is used. What time step size was used? This is a very old method, far more advance methods are available with implicit and automatic time steps.

We agree with the reviewer that newer and more advance methods are available (e.g., EK1 and EK0 [6]). However, implementation of new ODE solvers in Stan is not a straightforward task because of the need of their translation to Stan language [7]. Besides, we would like to emphasize that Stan makes use of the No-U-Turn Sampler (NUTS) to sample from a target distribution. As NUTS is based on the Hamiltonian Monte Carlo (HMC) algorithm [8]. Among other requirements, this implies a need of computing the gradient of the log-posterior by automatic differentiation before starting the sampler [9]. As a result, in the case of implementing an ODE solver with automatic time steps, it will make necessary to update the Jacobian matrix of the ODE model and hence, the log-posterior, at any time there is a change in step size leading to an unavoidable slowdown of computation time and also to a lengthy verification process for testing the adequacy of such a newer ODE solver (e.g., running tests and solving benchmark problems). Instead, we preferred to make use of the built-in Stan solver *rk45* that has been extensively used in a wide range of problems and applications (see Rosenbaum et al. [10], Bandiera et al. [11], Grinsztajn et al. [12] to name a few). We do also agree that it is necessary to specify the time step size used, reason why we have added in line 194 of our manuscript updated version, the following:

... with relative and absolute tolerance values of 1×10^{-6} for both, and a maximum number of steps of 1×10^4 .

4. L. 189: Why the burn in or warm up was left at 1,000 iterations? Why such a small sample size? (4 parallel chains of 3,000, why use parallel chains?).

We understand the concern of the reviewer in our apparently low number of iterations used for both, warm-up and sampling of the posterior distributions. Contrary to other sampling algorithms (e.g., Metropolis-Hastings and Gibbs), we here use a sampling, which explores the geometry of the posterior distribution in a more efficient way than purely random walk based approaches. This is due to its formulation on basis of the HMC algorithm [8, 9]. As a result, convergence using NUTS can be often reached after few thousand iterations, even for complex models [13]. About the use of 4 parallel chains, this decision is based on two main arguments: (1) running in parallel multiple chains with different starting values allows to check for convergence of independent chains once stationarity is reached (as it is usually represented by traceplots where the called “caterpillar” shape is accomplished) [14]. And, (2) four is the minimum number of multiple chains suggested for using MCMC-NUTS. This responds to the need of counting with multiple chains for diagnostic statistics (i.e., \hat{R} , tail effective sample size and bulk effective sample size) that perform better with multiple chains due to their nature of taking into account within and between chain variability [14, 15].

Note that this would be quite short for an independent sampler, let alone with the correlation inherent in the MCMC. What is then the effective sample size?

Another consequence of using NUTS is a considerable reduction in the correlation between drawn samples from the posterior distribution. Opposed to MCMC samplers based on random walks, due to its Hamiltonian dynamics, NUTS offers a more efficient exploration of the target distribution even in the presence of local correlations where the former samplers tend to get stuck [14, 16]. In consequence, adequate effective sample sizes (ESS) are obtained with less iterations. Thus, in response to the reviewer concern, we have added an extra table in the Supplementary Material as follows:

Table S6. Summary of effective sample size over total number of posterior draws (ESS/N_{draws}) among successful fits across 31 model iterations (MI). Average ESS/N_{draws} in the referred MI is reported. Column “MI()” indicates combination of mechanisms deployed in MI.

MI()	ghhp1	dowb1	ghhp2	ghhp3	ghhp4	ghhp5	brwb1	brwb2	ecpt1	ecpt2	ecwb1	ecwb2	brpb1	brst1	brwb3	brwb4	brst2	brwb7	mywb3	hnwb1	ecpb1	niwb1	niwb2
0	0.555	-	-	-	0.591	0.530	0.642	0.607	0.580	0.526	0.619	0.459	-	0.535	0.541	0.416	0.338	0.514	0.501	0.428	0.330	-	0.601
1	0.403	-	-	-	0.267	0.482	0.673	0.592	-	0.462	0.434	0.249	0.467	-	-	0.395	0.440	-	-	-	-	-	0.529
2	0.584	-	-	0.682	0.558	-	0.560	0.340	0.507	0.492	-	0.345	0.551	0.407	-	0.351	0.428	-	0.428	-	-	-	0.533
3	0.536	-	-	-	0.554	0.511	0.602	0.598	0.609	0.615	0.620	0.570	0.626	0.451	0.522	0.366	0.340	-	0.516	0.471	0.448	0.425	0.542
4	0.621	-	-	-	0.586	0.551	0.626	0.604	-	0.578	0.632	0.339	0.522	0.589	-	0.441	0.330	-	0.447	0.462	-	-	0.562
5	-	-	-	-	-	-	0.607	0.596	-	-	-	-	-	-	-	-	-	-	-	-	-	-	-
1,2	0.490	-	0.611	0.445	0.471	-	0.557	0.417	0.461	-	0.402	-	0.399	-	-	-	-	-	-	-	-	-	-
1,3	0.481	-	-	-	-	0.257	0.619	0.596	0.695	0.402	0.410	0.274	0.416	-	-	0.490	0.311	-	-	-	-	-	-
1,4	-	-	-	-	-	0.373	0.556	0.677	0.628	-	0.507	-	0.458	-	-	-	0.594	-	0.518	-	-	-	0.272
1,5	-	-	-	-	-	-	0.663	0.547	-	-	-	-	-	-	-	-	-	-	-	-	-	-	-
2,3	0.612	0.553	0.521	0.688	0.454	0.512	0.478	0.326	0.583	0.424	0.353	0.330	0.567	0.362	-	-	-	-	0.422	-	-	0.447	0.399
2,4	0.572	-	0.613	0.619	0.694	-	-	0.386	-	0.528	0.510	-	0.532	-	-	-	-	-	0.451	-	-	-	0.438
2,5	-	-	-	-	-	-	-	-	-	-	-	-	-	-	-	-	-	-	-	-	-	-	-
3,4	0.556	-	-	-	0.605	0.492	0.607	0.724	-	0.607	0.753	-	0.55	0.487	-	-	-	-	0.509	0.458	0.605	-	0.564
3,5	-	-	-	-	-	-	0.609	-	-	-	-	-	-	-	-	-	-	-	-	-	-	-	-
4,5	-	-	-	-	-	-	-	-	-	-	-	-	-	-	-	-	-	-	-	-	-	-	-
1,2,3	0.519	-	0.581	0.393	-	-	0.534	0.401	-	-	0.359	-	0.373	-	-	-	-	-	-	-	-	-	-
1,2,4	0.486	-	0.615	0.493	0.567	0.388	0.525	0.521	0.715	-	0.432	-	0.420	-	-	-	-	-	-	-	-	-	-
1,2,5	-	-	-	-	-	-	0.511	0.388	-	-	-	-	-	-	-	-	-	-	-	-	-	-	-
1,3,4	-	-	-	-	0.369	0.312	0.608	0.587	0.691	-	0.473	-	0.466	-	0.523	-	0.557	-	-	-	-	-	-
1,3,5	-	-	-	-	-	-	0.643	0.578	-	-	-	-	-	-	-	-	-	-	-	-	-	-	-
1,4,5	-	-	-	-	-	-	-	0.611	-	-	-	-	-	-	-	-	-	-	-	-	-	-	-
2,3,4	0.585	-	-	0.599	0.595	0.512	0.569	-	0.583	0.415	0.507	-	0.562	-	-	-	0.440	-	0.459	-	-	-	0.393
2,3,5	-	-	-	-	-	-	-	-	-	-	-	-	-	-	-	-	-	-	-	-	-	-	-
2,4,5	-	-	-	-	-	-	-	-	-	-	-	-	-	-	-	-	-	-	-	-	-	-	-
3,4,5	-	-	-	-	-	-	-	-	-	-	-	-	-	-	-	-	-	-	-	-	-	-	-
1,2,3,4	0.536	-	0.512	0.505	-	0.259	0.478	0.482	0.706	-	0.429	-	0.433	-	-	-	-	-	-	-	-	-	-
1,2,3,5	-	-	-	-	-	-	0.536	0.415	-	-	-	-	-	-	-	-	-	-	-	-	-	-	-
1,2,4,5	-	-	-	-	-	-	-	-	-	-	-	-	-	-	-	-	-	-	-	-	-	-	-
1,3,4,5	-	-	-	-	-	-	-	-	-	-	-	-	-	-	-	-	-	-	-	-	-	-	-
2,3,4,5	-	-	-	-	-	-	-	-	-	-	-	-	-	-	-	-	-	-	-	-	-	-	-
1,2,3,4,5	-	-	-	-	-	-	-	-	-	-	-	-	-	-	-	-	-	-	-	-	-	-	-

Where the averaged ratio of ESS with the total number of draws (ESS/N_{draws}) from the posteriors of each MI is presented. As a rule of thumb, it has been suggested that an advisable ESS should be in the thousands [15]. Having in mind that our approach used as basis 8000 iterations in total per each MI, in all successful MIs, ESS is acceptable. Consequently, we added at the end of Section 2.4.3 the following sentence:

Assessing autocorrelation of the sampled parameters was performed by means of averaging computed effective sample sizes over number of posterior draws (ESS/N_{draws}) and checking whether these were above 12.5%, meaning that as minimum 1000 ESS were obtained [15].

Finally, we added the following sentence in line 267 of the updated version of our manuscript, section 3.1:

Furthermore, in terms of autocorrelation, all successful fits showed averaged values of ESS/N_{draws} over 12.5% (see Supplementary Table S6), indicating an acceptable ESS above 1000.

5. Note that the corner stone of the paper is the model fit- model assessment. Section 2.5 is only very briefly explained and unclear. The authors should explain in far more detail their approach. For example, what is a “successful fit” (l. 199)?

We agree with the reviewer that we needed to be more detailed in defining what is a “successful fit”. With this in mind, we have added the following paragraph in Section 2.5:

Under the umbrella of the proposed Bayesian framework, it is important to define what will be called from now on a *successful fit*. For this aspect, we consider as such, any MI fit that converged according to the criteria described in Section 2.4.3 and does not involve any re-parametrization or use of distinct priors in cases where divergences of MCMC-NUTS or complete lack of sampling could arise as result of complicated geometries imposed to the posterior distributions by inclusion of the assessed mechanisms over particular datasets.

Note that there is a formal Bayesian approach to model adequacy-model selection: The posterior probability of each model is proportional to its normalization constant, see <https://doi.org/10.1214/ss/1009212519> for example. All other approaches are approximations to the latter (e.g. the BIC, DIC etc.). The problem is estimating such normalizations constants using MCMC, which remains a difficult problem, still, see <https://doi.org/10.1198/106186008X320258>. The authors should try to use this formal approach, before resorting to other more ad hoc or heuristic methods.

We have followed the reviewer’s suggestion of implementing Bayesian Model Averaging (BMA) [17]. For this purpose, we used the weights for each MI that resulted from performing pseudo-BMA [18] over sets of MIs that successfully fitted a given dataset and averaged them over the total number of datasets that were fitted at least once. Hence, we have modified Section 2.5 as follows:

On the one hand, Bayesian Model Averaging (BMA [17]) weights were computed using pseudo-BMA [18]. These weights, understood as representations of the relative probability of each MI [19], were then averaged over the total number of datasets that were fitted at least once by any MI, and used as a measure of the adequacy of each MI across all data. For computing the mean value of BMA weights (BMA_w), non-successful fits were assigned values of zero. Furthermore, an observed success rate (OSR) and expected success rate (ESR) were determined on the basis of times where the model was satisfactorily fit to a given dataset. OSR is then defined by the ratio of the number of successful fits and the total of datasets fitted by at least one MI. ESR, used to properly compare the success rates of models with only a single additional mechanism with those models containing a combination of mechanisms, is defined as the product of the OSRs of the elementary MIs. For model variant MI(1,2,5), for example, the expected success rate is then the product of the observed success rates of the elementary models MI(1), MI(2) and MI(5).

As a consequence, we have added in section 3.2 the following paragraphs:

Both approaches for assessing model success across all datasets, a formal one as BMA and our proposed measures OSR and ESR, showed similar conclusions over competing MIs.

From their computations, the baseline model model (MI(0)) showed the highest values ($BMA_w = 0.164$, $OSR = 0.78$). For MIs containing single mechanisms, M(1), M(2), M(3), M(4) and M(5), BMA_w were 0.084, 0.034, 0.075, 0.079 and 0.011, while OSR values were 0.52, 0.61, 0.83,

0.65 and 0.09, respectively. Among more complex combinations of mechanisms, BMA_w ranged between 0.00 to a maximum of 0.141 reached by the combination of M2 and M3 (MI(2,3)). Similarly, among these, MI(2,3) had a maximum OSR of 0.74 as listed in Table 4.

Furthermore, we updated Table 4 in the main manuscript accordingly:

Table 4. Summary of successful fits across 31 models iterations (MIs) and baseline. Light green-coloured cells indicate successful fits. Light-red coloured cells indicate non-successful fits. Columns “MI()”, “#”, “ BMA_w ”, “OSR” and “ESR” refer to combination of mechanisms deployed in model iteration, number of parameters, averaged Bayesian Model Averaging weights, observed success rate and expected success rate, respectively.

ghhp1	dowb1	ghhp2	ghhp3	ghhp4	ghhp5	brwb1	brwb2	ecpt1	ecpt2	ecwb1	ecwb2	brpb1	brst1	brwb3	brwb4	brst2	brwb7	mywb3	hnwb1	ecpb1	niwb1	niwb2	MI()	#	BMA_w	OSR	ESR	
																							0	24	0.164	0.78		
																								1	27	0.084	0.52	
																								2	31	0.034	0.61	
																								3	26	0.075	0.83	
																								4	28	0.079	0.65	
																								5	27	0.011	0.09	
																								1,2	34	0.021	0.39	0.32
																								1,3	29	0.048	0.48	0.43
																								1,4	31	0.045	0.39	0.34
																								1,5	30	0.008	0.09	0.05
																								2,3	33	0.141	0.74	0.5
																								2,4	35	0.031	0.43	0.4
																								2,5	34	0	0.04	0.05
																								3,4	30	0.066	0.57	0.54
																								3,5	29	0.002	0.04	0.07
																								4,5	31	0	0	0.06
																								1,2,3	36	0.006	0.3	0.26
																								1,2,4	38	0.017	0.43	0.21
																								1,2,5	37	0	0.09	0.03
																								1,3,4	33	0.052	0.39	0.28
																								1,3,5	32	0.001	0.09	0.04
																								1,4,5	34	0.001	0.04	0.03
																								2,3,4	37	0.100	0.52	0.33
																								2,3,5	36	0	0	0.04
																								2,4,5	38	0	0	0.03
																								3,4,5	33	0	0	0.05
																								1,2,3,4	40	0.003	0.39	0.17
																								1,2,3,5	39	0	0.09	0.02
																								1,2,4,5	41	0	0	0.02
																								1,3,4,5	36	0	0	0.02
																								2,3,4,5	40	0	0	0.03
																								1,2,3,4,5	43	0	0	0.01

Finally, in light of the results of BMA confirming better performance of some MIs involving mechanism 1 (M1), we have added in line 313 of the updated manuscript the following sentence:

The latter is also evident by higher BMA weights by such MIs (see Supplementary Table S7).

And added to the Supplementary Material, Table S7 as follows:

Table S7. Summary of Bayesian Model Averaging (BMA) among successful fits across 31 model iterations (MI). BMA's weights in the referred MI is reported. Column "MI()" indicates combination of mechanisms deployed in MI.

MI()	ghhp1	dowb1	ghhp2	ghhp3	ghhp4	ghhp5	brwb1	brwb2	ecpt1	ecpt2	ecwb1	ecwb2	brpb1	brst1	brwb3	brwb4	brst2	brwb7	mywb3	hnwb1	ecpb1	niwb1	niwb2
0	0.000	-	-	-	0.022	0.238	0.079	0.223	0.000	0.040	0.347	0.000	-	0.747	0.000	0.000	0.000	0.890	0.110	0.659	0.005	-	0.421
1	0.000	-	-	-	0.000	0.002	0.035	0.040	-	0.460	0.004	0.000	0.843	-	-	0.011	0.018	-	-	-	-	-	0.517
2	0.046	-	-	0.086	0.157	-	0.014	0.000	0.038	0.016	-	0.290	0.000	0.007	-	-	0.000	0.110	0.004	-	-	-	0.006
3	0.013	-	-	-	0.012	0.105	0.012	0.088	0.000	0.083	0.118	0.022	0.000	0.009	0.000	0.000	0.000	-	0.778	0.324	0.046	0.057	0.055
4	0.001	-	-	-	0.007	0.175	0.503	0.233	-	0.113	0.308	0.220	0.000	0.233	-	0.000	0.000	-	0.011	0.005	-	-	0.001
5	-	-	-	-	-	-	0.133	0.120	-	-	-	-	-	-	-	-	-	-	-	-	-	-	-
1,2	0.000	-	0.481	0.003	0.002	-	0.002	0.001	0.000	-	0.000	-	0.003	-	-	-	-	-	-	-	-	-	-
1,3	0.000	-	-	-	-	0.000	0.001	0.011	0.000	0.103	0.000	0.000	0.007	-	-	0.989	0.000	-	-	-	-	-	-
1,4	-	-	-	-	-	0.000	0.008	0.069	0.001	-	0.004	-	0.146	-	-	-	-	-	0.002	-	-	-	0.000
1,5	-	-	-	-	-	-	0.099	0.078	-	-	-	-	-	-	-	-	-	-	-	-	-	-	-
2,3	0.398	1.000	0.001	0.328	0.011	0.036	0.005	0.000	0.034	0.008	0.004	0.468	0.000	0.000	-	-	-	-	0.014	-	-	0.943	0.000
2,4	0.018	-	0.001	0.050	0.564	-	-	0.000	-	0.047	0.038	-	0.000	-	-	-	-	-	0.000	-	-	-	0.000
2,5	-	-	-	-	-	-	-	-	-	-	-	-	-	-	-	-	-	-	-	-	-	-	-
3,4	0.007	-	-	-	0.007	0.041	0.048	0.087	-	0.121	0.164	-	0.000	0.005	-	-	-	-	0.071	0.013	0.948	-	0.000
3,5	-	-	-	-	-	-	0.044	-	-	-	-	-	-	-	-	-	-	-	-	-	-	-	-
4,5	-	-	-	-	-	-	-	-	-	-	-	-	-	-	-	-	-	-	-	-	-	-	-
1,2,3	0.000	-	0.133	0.004	-	-	0.001	0.000	-	-	0.000	-	0.000	-	-	-	-	-	-	-	-	-	-
1,2,4	0.000	-	0.349	0.001	0.016	0.000	0.001	0.002	0.019	-	0.000	-	0.000	-	-	-	-	-	-	-	-	-	-
1,2,5	-	-	-	-	-	-	0.006	0.001	-	-	-	-	-	-	-	-	-	-	-	-	-	-	-
1,3,4	-	-	-	-	0.000	0.000	0.004	0.006	0.007	-	0.000	-	0.001	-	1.000	-	0.175	-	-	-	-	-	-
1,3,5	-	-	-	-	-	-	0.004	0.009	-	-	-	-	-	-	-	-	-	-	-	-	-	-	-
1,4,5	-	-	-	-	-	-	-	0.031	-	-	-	-	-	-	-	-	-	-	-	-	-	-	-
2,3,4	0.518	-	-	0.522	0.201	0.156	0.003	-	0.875	0.009	0.010	-	0.000	-	-	-	0.000	-	0.009	-	-	-	0.000
2,3,5	-	-	-	-	-	-	-	-	-	-	-	-	-	-	-	-	-	-	-	-	-	-	-
2,4,5	-	-	-	-	-	-	-	-	-	-	-	-	-	-	-	-	-	-	-	-	-	-	-
3,4,5	-	-	-	-	-	-	-	-	-	-	-	-	-	-	-	-	-	-	-	-	-	-	-
1,2,3,4	0.000	-	0.036	0.006	-	0.000	0.001	0.000	0.027	-	0.000	-	0.000	-	-	-	-	-	-	-	-	-	-
1,2,3,5	-	-	-	-	-	-	0.000	0.000	-	-	-	-	-	-	-	-	-	-	-	-	-	-	-
1,2,4,5	-	-	-	-	-	-	-	-	-	-	-	-	-	-	-	-	-	-	-	-	-	-	-
1,3,4,5	-	-	-	-	-	-	-	-	-	-	-	-	-	-	-	-	-	-	-	-	-	-	-
2,3,4,5	-	-	-	-	-	-	-	-	-	-	-	-	-	-	-	-	-	-	-	-	-	-	-
1,2,3,4,5	-	-	-	-	-	-	-	-	-	-	-	-	-	-	-	-	-	-	-	-	-	-	-

6. Please explain in more detail Section 2.6. Why the PCA is done? What is its purpose? It uses all data sets and all models also or only the winner model? Can the authors use the uncertainty in the posterior and not only use the medians (a point estimate)?

We have followed the reviewer recommendation and added a more detailed description of the PCA conduction. Hence, in section 2.6 we have added the following sentence:

In an approach of linking our mathematical exploration of mechanisms with a real-life application (where Principal Component Analysis (PCA) is a common approach), we studied, whether projecting the obtained vectors of kinetic parameters on the PCA space allows for a separation (and hence classification) of experiments according to fermentation features (e.g., cocoa beans' country of origin and used cultivar), which are of interest in both, academic research and chocolate industry.

Regarding the reviewers' concern of whether we did use all data sets and all models for performing PCA, we indeed did it, and we have pointed out this by adding the following sentence in Section 2.6:

Consequently, six main features of fermentation trials were taken into account as groups and analysed via PCA over parameter estimates for all successful MIs.

Concerning the possibility of using the uncertainty in the posterior for conducting PCAs, we have followed the recommendation of the reviewer. For accomplishing it, we have used all posterior draws within the 95% credible interval (CI), as now stated in Section 2.6 in the following sentence:

Only the posterior samples contained within their 95% credible interval (CI) from each chain in the MCMC-NUTS runs were taken into account to perform PCA.

As one could expect, by using the uncertainty of the posteriors for PCAs, led to a change in the reported explained variances that we have updated in the following manner:

Line 343 ... with explained variances of 13.91% by the first PCA component (PC1) and 9.85% by the second component (PC2).

Line 352: ... with PC1 and PC2 scores explaining 11.87% and 8.76% of variance ...

Line 359: ... with total explained variance of 16.61% by PC1 and 9.97% by PC2 ...

Line 372: ... explains a total variance of 45.8% for PC1 and 25.89% for PC2.

It is important to mention that conducting PCAs in this way, has not drastically changed our original results and interpretations. We have added minor corrections in light of the updated PCAs as follows:

(1) In Section 3.4.4, we originally reported a clear separation for all countries in the PCA scores. From this new version, we have found that parameters scores for Ecuador and Ghana overlap between each other (see updated Supplementary Figure S29 below), reason why we have rephrased the sentence in line 372 as follows:

Furthermore, a clear separation between Brazil with respect to Ecuador and Ghana, is defined over PC1.

(2) The obtained scores and loadings from PCAs in this updated version have been projected mirroring our original findings for features “fermentation method” and “country of origin” (see updated Supplementary Figures S28 and S29 below), which led us to modify the following sentences:

Line 360: Furthermore, parameters with positive loadings in PC1, namely ...

Line 373: From its loadings, parameters with positive loadings in PC1 ...

Consequently, figures corresponding to the PCA scores and loadings have been updated. Specifically Figures 3 and 4 of the main manuscript, and Figures S28 and S29 in the supplementary material.

Figure 3. PCA score (a) and loading plot (b) from all parameters of model iteration MI(2,3), feature cultivar. For visualization purposes, scores of only 10% of posterior draws are shown. Criollo/Forastero (cf), Nacional/Trinitario (nt), unknown cultivar used by Camu et al. [20] (un1) and unknown cultivar used by Pereira et al. [21] (un2) are shown. Parameters located on the left and right with respect to 0 in PC1 loading plot determine differentiation between cf, nt and un2.

Figure 4. PCA score (a) and loading plot (b) from all parameters of model iteration MI(2,3), feature temperature. For visualization purposes, scores of only 10% of posterior draws are shown. Controlled temperature (ctrl) and non-controlled temperature (nctrl) are shown. Parameters located on the left and right with respect to 0 in PC1 loading plot determine differentiation between ctrl and nctrl.

Figure S28. PCA score (a) and loading plot (b) from all parameters of model iteration MI(2). For visualization purposes, scores of only 10% of posterior draws are shown. Heap (hp), platform (pt), stainless-steel tank (st) and wooden box (wb) fermentation methods are shown. Parameters located on the left and right with respect to 0 in the horizontal axis in the loading plot determine differentiation between stainless-steel tank and rest of fermentation methods, respectively.

Figure S29. PCA score (a) and loading plot (b) from lactic acid bacteria-related parameters of model iteration M1(1,3,4). For visualization purposes, scores of only 10% of posterior draws are shown. Brazil (BR) is clearly separated from Ecuador (EC) and Ghana (GH). Parameters located on the left and right with respect to 0 in the horizontal axis in the loading plot determine differentiation between all classes.

Finally, we have updated also Figure S27 regarding Mahalanobis distances, which has not changed our original findings either.

Figure S27. Heat map of medians of pairwise squared Mahalanobis distances (\tilde{D}_M) between centroids of grouping classes scores from Principal Component Analysis (PCA) per model iteration and subgroups of parameter estimates. (a) Country, (b) cocoa cultivar, (c) fermentation method, (d) turning of fermenting mass, and (e) use of controlled temperature. White rows in (b), (c) and (e) correspond to model iterations where only one group class was available and PCA groupings could not be performed. Subgroups ALL, MSGR, MR, YC, Y-related, LAB-related and AAB-related correspond to all common MIs parameter estimates, maximum specific growth rates, yield coefficients, yeast-related parameters, lactic acid bacteria-related parameters, and acetic acid bacteria-related parameters, respectively.

7. The authors seem to trust nearly blindly the statistics \hat{R} etc. The authors are wrong here in believing that a bad convergence in the MCMC means a bad model. The model can be very good but results in a difficult posterior to sample from and the MCMC method itself is what is inadequate. The U-turn sampler is not bullet proof, nor any of the other methods programmed in Stan. Difficult posteriors (multimodal, too correlated) are difficult to sample for any method and the methods in Stan are no exceptions (btw contrary to what the authors of Stan seem to claim). This misconception is also apparent in the Discussion, l. 354. On the contrary, the posterior may be very simple (nice unimodal bell shape) to sample from, the MCMC be excellent but the model itself be useless.

Conceptually, the authors need to separate MCMC convergence and model fit. The former needs to be assured first, if no convergence is attained, a larger sample size may be needed or a different sampler altogether. There are other multipurpose samplers that may be used, like DRAM or the t-walk MCMC samplers. Once MCMC convergence has been attained then model adequacy may be analyzed.

I am confident that the authors results will hold, in general, since a detailed process of model assessment was done, but a more methodologically and conceptually sound analysis is in order.

We agree with the reviewer in that we did not make a clear definition between bad convergence and model fitting. After our corrections, specially the one given as response for comment 5 of the reviewer, we have made clear a definition of what it is considered a successful fit under our Bayesian framework. Taking such a definition as a starting point, we now feel confident that the structure of our manuscript has a well defined final separation between MCMC convergence and model fit by changing “convergence” wording for more appropriate phrases in line with our definition of successful fits in the Discussion section of our manuscript. In more detail, these changes in our updated version are:

Line 378: Section 4.1 has been renamed as “Model plausibility”.

Lines 380, 425, 435: Change of the word “convergence” for “success”, “sample from complicated geometries of the posterior target” and “successful fits” respectively.

Lines 381, 385, 393, 396, 413, 449: Change of the word “non-convergence” by “failure in fitting” in line 381, and “fitting failure” in the rest of the cases.

Finally, we would like to thank the anonymous reviewer for their advise that has led us to an improved new version of our original manuscript.

References

1. Mauricio Moreno-Zambrano, Sergio Grimbs, Matthias S. Ullrich, and Marc-Thorsten Hütt. A mathematical model of cocoa bean fermentation. *Royal Society Open Science*, 5(10):180964, 2018. doi: 10.1098/rsos.180964.
2. Nicholas Camu, Tom De Winter, Kristof Verbrugghe, Ilse Cleenwerck, Peter Vandamme, Jemmy S. Takrama, Marc Vancanneyt, and Luc De Vuyst. Dynamics and biodiversity of populations of lactic acid bacteria and acetic acid bacteria involved in spontaneous heap fermentation of cocoa beans in Ghana. *Applied and Environmental Microbiology*, 73(6):1809–1824, 2007. doi: 10.1128/AEM.02189-06.
3. Jonah Gabry, Daniel Simpson, Aki Vehtari, Michael Betancourt, and Andrew Gelman. Visualization in Bayesian workflow. *Journal of the Royal Statistical Society: Series A (Statistics in Society)*, 182(2): 389–402, 2019. doi: 10.1111/rssa.12378.
4. Andrew Gelman, Jessica Hwang, and Aki Vehtari. Understanding predictive information criteria for Bayesian models. *Statistics and Computing*, 24(6):997–1016, Nov 2014. doi: 10.1007/s11222-013-9416-2.

5. Rodney W. Strachan and Herman K. van Dijk. Bayesian model selection with an uninformative prior*. *Oxford Bulletin of Economics and Statistics*, 65(s1):863–876, 2003. doi: 10.1046/j.0305-9049.2003.00095.x.
6. Nathanael Bosch, Philipp Hennig, and Filip Tronarp. Calibrated Adaptive Probabilistic ODE Solvers. In Arindam Banerjee and Kenji Fukumizu, editors, *Proceedings of The 24th International Conference on Artificial Intelligence and Statistics*, volume 130 of *Proceedings of Machine Learning Research*, pages 3466–3474. PMLR, 13–15 Apr 2021.
7. W. R. Gillespie. Bayesian pharmacometric modeling with BUGS, NONMEM and Stan. 2015. URL <https://www.bayes-pharma.org/wp-content/uploads/2014/10/BayesianPmetricsBAYES2015.pdf>. Accessed: 30.08.2021.
8. Simon Duane, A.D. Kennedy, Brian J. Pendleton, and Duncan Roweth. Hybrid Monte Carlo. *Physics Letters B*, 195(2):216–222, 1987. doi: 10.1016/0370-2693(87)91197-X.
9. Matthew D. Hoffman and Andrew Gelman. The No-U-turn Sampler: Adaptively Setting Path Lengths in Hamiltonian Monte Carlo. *Journal of Machine Learning Research*, 15(1):1593–1623, January 2014.
10. Benjamin Rosenbaum, Michael Raatz, Guntram Weithoff, Gregor F. Fussmann, and Ursula Gaedke. Estimating parameters from multiple time series of population dynamics using bayesian inference. *Frontiers in Ecology and Evolution*, 6:234, 2019. ISSN 2296-701X. doi: 10.3389/fevo.2018.00234.
11. L. Bandiera, D. Gomez Cabeza, E. Balsa-Canto, and F. Menolascina. Bayesian model selection in synthetic biology: factor levels and observation functions. *IFAC-PapersOnLine*, 52(26):24–31, 2019. ISSN 2405-8963. doi: 10.1016/j.ifacol.2019.12.231. 8th Conference on Foundations of Systems Biology in Engineering FOSBE 2019.
12. Léo Grinsztajn, Elizaveta Semenova, Charles C. Margossian, and Julien Riou. Bayesian workflow for disease transmission modeling in Stan, 2021. Preprint.
13. Paul-Christian Bürkner. brms: An R Package for Bayesian Multilevel Models Using Stan. *Journal of Statistical Software, Articles*, 80(1):1–28, 2017. doi: 10.18637/jss.v080.i01.
14. Aki Vehtari, Andrew Gelman, Daniel Simpson, Bob Carpenter, and Paul-Christian Bürkner. Rank-Normalization, Folding, and Localization: An Improved \hat{R} for Assessing Convergence of MCMC. *Bayesian Analysis*, Jul 2020. doi: 10.1214/20-ba1221.
15. Andrew Gelman, Aki Vehtari, Daniel Simpson, Charles C. Margossian, Bob Carpenter, Yuling Yao, Lauren Kennedy, Jonah Gabry, Paul-Christian Bürkner, and Martin Modrák. Bayesian Workflow, 2020. Preprint.
16. M. Betancourt and M. Girolami. Hamiltonian Monte Carlo for Hierarchical Models. In S. K. Upadhyay, U. Singh, D. K. Dey, and A. Loganathan, editors, *Current Trends in Bayesian Methodology with Applications*, chapter 4, pages 79–95. Chapman & Hall/CRC, New York, 1st edition, 2015.
17. Jennifer A. Hoeting, David Madigan, Adrian E. Raftery, and Chris T. Volinsky. Bayesian model averaging: a tutorial. *Statistical Science*, 14(4):382 – 417, 1999. doi: 10.1214/ss/1009212519.
18. Yuling Yao, Aki Vehtari, Daniel Simpson, and Andrew Gelman. Using Stacking to Average Bayesian Predictive Distributions (with Discussion). *Bayesian Analysis*, 13(3):917 – 1007, 2018. doi: 10.1214/17-BA1091.
19. Carsten F. Dormann, Justin M. Calabrese, Gurutzeta Guillera-Arroita, Eleni Matechou, Volker Bahn, Kamil Bartoń, Colin M. Beale, Simone Ciuti, Jane Elith, Katharina Gerstner, Jérôme Guelat, Petr Keil, José J. Lahoz-Monfort, Laura J. Pollock, Björn Reineking, David R. Roberts, Boris Schröder, Wilfried Thuiller,

David I. Warton, Brendan A. Wintle, Simon N. Wood, Rafael O. Wüest, and Florian Hartig. Model averaging in ecology: a review of Bayesian, information-theoretic, and tactical approaches for predictive inference. *Ecological Monographs*, 88(4):485–504, 2018. doi: 10.1002/ecm.1309.

20. Nicholas Camu, Ángel González, Tom De Winter, Ann Van Schoor, Katrien De Bruyne, Peter Vandamme, Jemmy S. Takrama, Solomon K. Addo, and Luc De Vuyst. Influence of Turning and Environmental Contamination on the Dynamics of Populations of Lactic Acid and Acetic Acid Bacteria Involved in Spontaneous Cocoa Bean Heap Fermentation in Ghana. *Applied and Environmental Microbiology*, 74(1): 86–98, 2008. doi: 10.1128/AEM.01512-07.
21. Gilberto Vinícius de Melo Pereira, Maria Gabriela da Cruz Pedrozo Miguel, Cíntia Lacerda Ramos, and Rosane Freitas Schwan. Microbiological and physicochemical characterization of small-scale cocoa fermentations and screening of yeast and bacterial strains to develop a defined starter culture. *Applied and Environmental Microbiology*, 78(15):5395–5405, 2012. doi: 10.1128/AEM.01144-12.

Appendix C

Response to Reviewer's Comments on the Revision of the Manuscript

Exploring cocoa bean fermentation mechanisms by kinetic modelling

Mauricio Moreno-Zambrano¹, Matthias S. Ullrich¹, and Marc-Thorsten Hütt¹

¹Department of Life Sciences and Chemistry, Jacobs University Bremen, Campus Ring 1, 28759 Bremen, Germany

Dear Editors,

We thank the reviewers for their comments and recommendations on previous versions of our manuscript. Undoubtedly, they contributed to enhancing its quality. Here, you will find a detailed response to the remaining comments, leading to the enclosed revised version of our manuscript.

The original reviews are reproduced in black font, while our response is given in blue font.

Mauricio Moreno-Zambrano (for the authors)

Reviewer 1:

Comments to the Author(s) This revised version of the paper addresses correctly the concerns of the original review. In my opinion, the paper is ready for publication. I have a few additional minor comments:

There is still a problem with considering that “few thousand iterations” is sufficient for “convergence”. Note that even with independent MC sampling, one needs to take into consideration the variance of the statistic being approximated. The use of magic numbers like 1,000 or 100,000 samples is widespread but is simply wrong.

In order to clarify the reviewer’s concern, we have added Section 4 in the Supplementary Material as follows:

4 Convergence Criteria

A deeper understanding of the convergence criteria in this work is explained considering two main points: (1) convergence statistics used, and (2) how these statistics depend on the number of iterations and efficiency of the MCMC sampler.

First point: Our research was conducted using convergence statistics and criteria as proposed by Vehtari *et al.* [1], where the potential scale reduction factor \widehat{R} corresponds to an improved version of the split- \widehat{R} [2]. This procedure relies on a rank-normalized effective sample size (ESS) that ensures obtaining reliable estimates of variances and autocorrelations needed for its computation. Additionally, this allows to improve convergence diagnosis by the improved \widehat{R} in presence of heavy tails by introducing two new ESS measures, namely tail-ESS and bulk-ESS. However, for assessing convergence it is not enough to rely on a small value for \widehat{R} as it can occur that reasonable values of this statistic could be estimated along with poor ESS, specially tail-ESS and bulk-ESS. These last two are important in checking on the stability of the variances of the estimates given they represent specifically the ESS for extreme quantiles (e.g., 5% and 95% quantiles of the posterior). Second point: As is well known, MCMC methods need to be run long enough to accomplish convergence and obtain reliable uncertainties of the estimates. In this sense, increasing the number of iterations is among the most common practices to asses convergence through the examination of \widehat{R} and ESS. Roughly speaking, larger iterations help by regularizing the variance of the chains and decreasing autocorrelation of the samples. However, the question of how long the chains in any MCMC algorithm should be, does not have a universal answer as it depends highly on the efficiency of the sampler in exploring the parameter space [3]. Usually, classic MCMC algorithms (e.g. Metropolis-Hastings and Gibbs sampler) require quite high number of iterations to obtain reliable uncertainties because of the autocorrelation of their samples which are the result of their inherent random walk nature that governs them. In contrast, newer algorithms based on Hamiltonian dynamics as the MCMC No-U-Turn sampler have satisfactorily shown better efficiency than Metropolis-Hastings and Gibbs sampler in exploring the parameter space [4, 5]. As a result, assessing convergence of a model can require a considerable less amount of iterations, as long as, \widehat{R} and the ESS are within reasonable thresholds. In practical terms of our research, all aforementioned statistics were considered according to their suggested thresholds of 1.05 for \widehat{R} [6], 1000 for ESS [7], and 400 for both, tail-ESS and bulk-ESS [1], which consequently resulted in relative small number of iterations.

Addressing its inclusion in the main manuscript by adding at line 203 of its corrected version the following:

... (a more detailed explanation on the convergence criteria is described in Supplementary Material, Section 4).

In conclusion, we are certain that our chosen number of iterations for fitting each model is in line to what specialized literature suggest as good practices for assessing convergence.

I'm ok with the authors a practical approach to defining successful fit for which the MCMC-NUTS did not fail. With the new wording, this seems reasonable. However, please clarify that this is due to the limitations of the MCMC-NUTS: The authors of Stan would like to make us believe that if their sampler cannot handle a posterior distribution, then the user's posterior is wrong and the user most change their priors, models etc. This is a ridiculous and very convenient argument by the authors of Stan to justify their questionable sampler (which the authors of this paper are all very keen to defend).

We agree with the reviewer that limitations of MCMC-NUTS are also among possible causes for our models to fail. To address this concern, we have added among the possible causes for their failure in line 384 of the Discussion, section 4.1 of the corrected manuscript the following:

... , and even due to known limitations of MCMC-NUTS, such as poor chain mixing in presence of multimodal posteriors [8] and its higher sensitivity to model parameterization with respect to other samplers [9].

It will be very illustrating to now mention in the paper that, regarding the PCA analysis, inclusion of the uncertainty in posterior, using the MCMC draws, does make a difference in the analysis.

Following the reviewer's recommendation, we have added an extra paragraph in line 506 of the Discussion, section 4.3 of the corrected manuscript as follows:

Lastly, it is worth to remark that including all parameters' posterior draws contained in their 95% CI for conducting PCAs plays an important role in finding more meaningful differences between the features. This conclusion comes from a former approach were we performed PCAs over the medians of each chain of successful MIs instead. In such an exercise, besides obvious quantitative differences between its PCs and their counterparts reported here (see Supplementary Table S9), the same classification of the features was observed, with the exception of a noticeable separation between all countries (see Supplementary Figure S31) rather than the overlap between Ghana and Ecuador shown in Supplementary Figure S29, which could have been ignored if it were not by considering the posterior distributions' uncertainties.

Consequently, we have added Supplementary Table S9 in the supplementary material:

Table S9. Explained variance (EV) by the first two principal components of PCAs performed with medians and samples contained within 95% credible intervals (CI) of the posterior distributions. Columns EV_1 and EV_2 show EV for PCAs performed over MIs reported in Section 3.4MM using medians and 95% CI, respectively. EV is expressed as percentage.

Feature	EV_1	EV_2
Cultivar	48.36	23.76
Temperature	41.11	20.63
Method	51.23	26.58
Country	89.82	71.69

and Supplementary Figure S31:

Figure S31. PCA score (a) and loading plot (b) from lactic acid bacteria-related parameters of model iteration MI(1,3,4) using medians of posterior distributions. All countries, Brazil (BR), Ecuador (EC) and Ghana (GH) are clearly separated. Parameters located on the left and right with respect to 0 in the horizontal axis in the loading plot determine differentiation between all classes.

Finally, we would like to thank again the anonymous reviewer for their advise which had led us to improve this final version of our manuscript.

References

1. Vehtari, A., Gelman, A., Simpson, D., Carpenter, B. & Bürkner, P.-C. 2021 Rank-Normalization, Folding, and Localization: An Improved \hat{R} for Assessing Convergence of MCMC (with Discussion). *Bayesian Analysis*, **16**(2), 667 – 718. (doi:10.1214/20-BA1221)
2. Gelman, A., Carlin, J., Stern, H., Dunson, D., Vehtari, A. & Rubin, D. 2013 *Bayesian Data Analysis*. Chapman and Hall/CRC, 3rd edn.
3. Robert, C. P., Elvira, V., Tawn, N. & Wu, C. 2018 Accelerating MCMC algorithms. *WIREs Computational Statistics*, **10**(5), e1435. (doi:10.1002/wics.1435)
4. Hoffman, M. D. & Gelman, A. 2014 The No-U-turn Sampler: Adaptively Setting Path Lengths in Hamiltonian Monte Carlo. *Journal of Machine Learning Research*, **15**(1), 1593–1623.
5. Betancourt, M. & Girolami, M. 2015 Hamiltonian Monte Carlo for Hierarchical Models. In *Current Trends in Bayesian Methodology with Applications* (eds S. K. Upadhyay, U. Singh, D. K. Dey & A. Loganathan), chap. 4, pp. 79–95. New York: Chapman & Hall/CRC, 1st edn.
6. Stan Development Team 2020 RStan: the R interface to Stan. R package version 2.21.1.
7. Gelman, A., Vehtari, A., Simpson, D., Margossian, C. C., Carpenter, B., Yao, Y., Kennedy, L., Gabry, J., Bürkner, P.-C. *et al.* 2020 Bayesian Workflow. ArXiv preprint arXiv:2011.01808.

8. Mangoubi, O., Pillai, N. S. & Smith, A. 2018 Does Hamiltonian Monte Carlo mix faster than a random walk on multimodal densities? ArXiv preprint arXiv:1808.03230.
9. Monnahan, C. C., Thorson, J. T. & Branch, T. A. 2017 Faster estimation of Bayesian models in ecology using Hamiltonian Monte Carlo. *Methods in Ecology and Evolution*, **8**(3), 339–348. (doi: 10.1111/2041-210X.12681)